# Look Twice Before You Answer: Memory-Space Visual Retracing for Hallucination Mitigation in Multimodal Large Language Models

**Xin Zou** [* 1 2] **Yizhou Wang** [* 1] **Yibo Yan** [1 3] **Yuanhuiyi lyu** [1 3] **Kening Zheng** [1] **Sirui Huang** [1 4] **Junkai Chen** [1] **Peijie Jiang** [2] **Jia Liu** [2] **Chang Tang** [5] **Xuming Hu** [1 3]

## Abstract

Despite their impressive capabilities, multimodal large language models (MLLMs) are prone to hallucinations, *i.e.*, the generated content that is nonsensical or unfaithful to input sources. Unlike in LLMs, hallucinations in MLLMs often stem from the sensitivity of text decoder to visual tokens, leading to a phenomenon akin to "amnesia" about visual information. To address this issue, we propose MemVR, a novel decoding paradigm inspired by common cognition: *when the memory of an image seen the moment before is forgotten, people will look at it again for factual answers*. Following this principle, we treat visual tokens as supplementary evidence, re-injecting them into the MLLM through Feed Forward Network (FFN) as "key-value memory" at the middle trigger layer. This *look-twice* mechanism occurs when the model exhibits high uncertainty during inference, effectively enhancing factual alignment. Comprehensive experimental evaluations demonstrate that MemVR significantly mitigates hallucination across various MLLMs and excels in general benchmarks without incurring additional time overhead. The implementation is available from https://github.com/1zhou-Wang/MemVR.

## 1 Introduction

Multimodal Large Language Models (MLLMs), known for their ability to process visual, auditory and textual data, are crucial in fields such as computer vision (Koh et al., 2024) and natural language processing (Tu et al., 2023), helping

with visual tasks and complex visual question answering. However, MLLMs still face challenges, notably the "hallucination" issue (Huang et al., 2024b; Zheng et al., 2024; Lyu et al., 2025), where they generate contents inconsistent with original inputs, such as generating nonexistent objects or conflicting judgments. This flaw undermines their reliability, especially in areas critical to safety such as healthcare (Lin et al., 2024) and autonomous driving (Ding et al., 2024). Although the causes of hallucinations are unclear, one potential factor is the imbalance between their understanding of visual and textual modalities. This imbalance may induce biases when the model integrates multimodal information, leading to outputs that do not match objective facts.

Currently, numerous methods are being tried out to solve this problem. General studies can be broadly categorized into four streams: (i) Retrieval-Augmented Generation (RAG) (Qu et al., 2024) which incorporates knowledge from external databases to mitigate hallucinations, as well as (ii) through extra fine-tuning (Yu et al., 2024a) to enhance the self-consistency of generation; (iii) attention intervention (Huang et al., 2024a) and (iv) Contrastive Decoding (CD) (Leng et al., 2024) strategies, which not involve extra training. Specifically, RAG and fine-tuning patterns typically employ external knowledge retrieval or robust instruction-tuning datasets to post-hoc debias (Yang et al., 2024; Liu et al., 2023a), which inevitably introduces substantial computational overhead or storage requirements. Attention intervention, though not requiring additional data, usually involves retrospection-allocation operations, which bring about high inference latency and a large memory footprint.

CD-based methods (Li et al., 2023a; Shi et al., 2024) represent a simpler and more efficient way to mitigate hallucinations than other paradigms. Specifically, CD-based hallucination mitigation paradigms, represented by VCD (Leng et al., 2024), modulate logits of the next token prediction in a contrastive manner. As shown in Figure 1 left, VCD amplifies the language priors by adding Gaussian noise to the visual inputs, reducing over-reliance on statistical biases and single-modal priors through contrasting output distributions from original and distorted visual inputs. This perturbation of original inputs requires task-specific design, inevitably

---

[*]Equal contribution [1]The Hong Kong University of Science and Technology (Guangzhou) [2]Ant Group [3]The Hong Kong University of Science and Technology [4]University of Technology Sydney [5]Huazhong University of Science and Technology. Correspondence to: Xuming Hu <xuminghu@hkust-gz.edu.cn>.

*Proceedings of the 42nd International Conference on Machine Learning*, Vancouver, Canada. PMLR 267, 2025. Copyright 2025 by the author(s).

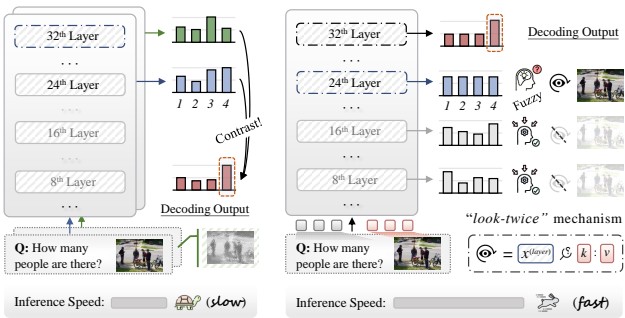

Figure 1: Comparison of the conventional CD-based hallucination mitigation paradigm VCD, and our proposed efficient MemVR.

✄ doubling inference costs. More critically, contrastive distributions are agnostic to visual and instructional nuances, which may not always amplify the intended hallucinations, occasionally ✄ introducing potential noise into CD.

In this work, we delve into the challenges of hallucination mitigation in MLLMs and propose a novel paradigm that overcomes the two limitations of CD-based approaches. Our research is grounded in a common cognitive process: *when the memory of an image seen the moment before is forgotten, it is intuitive to look twice for accurate and factual answers* (Ballard et al., 1995; Horowitz & Wolfe, 1998). Following this principle, we propose Memory-space Visual Retracing (MemVR) that mitigates hallucinations through supplementing visual evidence, which can also be called *look-twice* mechanism. As shown in Figure 1 right, MemVR reinjects visual tokens through Visual Retracing (VR), *i.e.*, *look-twice* mechanism, into the middle trigger layer suffering from high uncertainty, without incurring additional inference cost. Compared with VCD and other approaches, our proposed *look-twice* mechanism is optimal in terms of performance, efficiency, and memory cost as shown in Figure 2 and Table 1. Through extensive experiments on multimodal hallucination benchmarks, as well as GPT-4o evaluations, including eight public benchmarks, we show the comprehensive performance improvements of MemVR in hallucination mitigation and general capabilities. The main contributions can be summarized as follows:

❶ We propose MemVR, a novel, efficient, minimalist, and plug-and-play approach that achieves both model fidelity and efficiency, which reinforces attention to visual information for enhancing modality balance during the forward pass, without eliminating beneficial language priors.

❷ We present static and dynamic VR strategies that shift hidden states of the intermediate layer in MLLM for self-enhancement, rather than modulating logits directly in a CD manner, thus avoiding multi-round decoding.

❸ Our analysis reveals that hallucinations are triggered by the sensitivity of text decoder (*i.e.*, LLM) to non-text modality. This finding is experimentally validated.

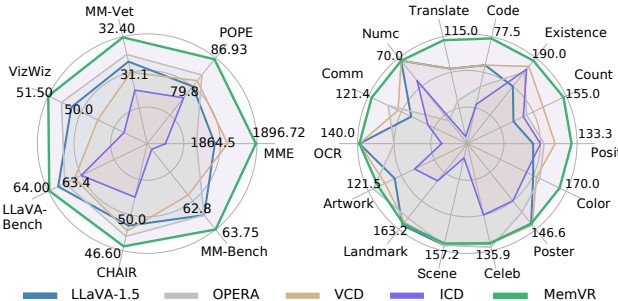

Figure 2: Radar charts comparing models across benchmarks.

| Method | Latency ↓ (ms/token) | Throughput ↑ (token/ms) | Time Cost ↓ for 80 Tokens (ms) | Memory ↓ Cost (MB) |
|---|---|---|---|---|
| Greedy | 65.71 (×1.00) | 0.015 (×1.00) | 5256.6 (×1.00) | 14257 (×1.00) |
| Sample | 69.84 (×1.06) | 0.015 (×1.00) | 5587.0 (×1.00) | 14262 (×1.00) |
| OPERA | 240.59 (×3.66) | 0.004 (×0.27) | 19247.2 (×3.66) | 21300 (×1.49) |
| ICD | 70.84 (×1.08) | 0.014 (×0.93) | 5666.9 (×1.08) | 14263 (×1.00) |
| VCD | 144.62 (×2.20) | 0.007 (×0.47) | 11569.3 (×2.20) | 14967 (×1.05) |
| MemVR | 68.32 (×1.04) | 0.015 (×1.00) | 5545.5 (×1.06) | 14345 (×1.01) |

Table 1: Performance comparison of SOTA methods and MemVR in latency, throughput, time cost, and memory usage. The best and suboptimal results are highlighted in green and blue, respectively.

❹ Comprehensive experiments and evaluations on multiple models demonstrate that MemVR outperforms SOTA methods in performance and inference speed, such as +7.0% on the POPE benchmark, 15.6% improvement on CHAIR$_I$, and total score of MME increased by +32.2 marks. The results on popular hallucination and general benchmarks validate the generalizability of our method.

## 2 Related Work

**MLLMs and Challenges.** In recent years, MLLMs have made remarkable progress, particularly as they have evolved from the foundations laid by Vision Language Models (VLMs). Early based on BERT-style language decoders (Devlin, 2018), which achieved initial cross-modal integration by combining visual and textual data (Li et al., 2022). Leveraging open-source Large Language Models (LLMs) such as LLaMA families (Touvron et al., 2023), MLLMs (Alayrac et al., 2022; Wu et al., 2024) have demonstrated enhanced adaptability across a range of visual language tasks, leading to a more profound ability to interpret the world. Models like LLaVA (Liu et al., 2024b), Qwen-VL (Bai et al., 2023), GLM4V (Wang et al., 2023), and LLaVA-Next (Liu et al., 2024a) have further advanced this field, enabling users to interact with these agents using both image and text prompts. These models adhere to two critical training phases: pre-training feature alignment and instruction fine-tuning, ensuring they better comprehend the format of instruction inputs (Yin et al., 2024). However, despite their impressive performance in many areas, multimodal large language models still suffer from hallucination issues. In this work, we conducted experiments and analysis on these representative models to validate MemVR's applicability.

Table 2: Comprehensive comparisons between the proposed MemVR and existing approaches are presented. MemVR introduces a low-latency *look-twice* decoding mechanism, optimizing hidden states to support multimodal integration and enhance overall performance. MemVR uniquely achieves visual hallucination mitigation and general improvement. SOTA methods we compare are emphasized in gray.

| Methods | VH Mitigation | General Improvement | Inference Latency | Expand to More Modalities | Modified Component(s) | Decoding Paradigm |
|---|---|---|---|---|---|---|
| DoLa (Chuang et al., 2023) | Negative | ✗ | Low | ✓ | Logits | Contrastive |
| OPERA (Huang et al., 2024a) | Medium | ✗ | High | ✓ | Attention matrix | Att-intervention |
| EAH (Zhang et al., 2024a) | Medium | ✗ | High | ✓ | Attention matrix | Att-intervention |
| CCA (Xing et al., 2024) | Medium | ✗ | High | ✓ | Attention matrix | Att-intervention |
| ICD (Wang et al., 2024) | Medium | ✗ | Medium | ✗ | Textual input, logits | Contrastive |
| ID (Kim et al., 2024) | Medium | ✗ | Medium | ✗ | Textual input, logits | Contrastive |
| SID (Huo et al., 2024) | Medium | ✗ | Medium | ✗ | Visual tokens, logits | Contrastive |
| VCD (Leng et al., 2024) | Low | ✗ | Medium | ✗ | Visual input, logits | Contrastive |
| HALC (Chen et al., 2024b) | Medium | ✗ | Ultrahigh | ✗ | Visual input, logits | Contrastive |
| VORD (Neo & Chen, 2024) | Medium | ✗ | Medium | ✗ | Visual input, logits | Contrastive |
| MemVR (ours) | High | ✓ | Low | ✓ | Hidden states | Look-twice |

**Mitigating Hallucinations in MLLMs.** Researchers have made extensive efforts to uncover the causes of hallucinations (Yin et al., 2023; Zhou et al., 2023; Bai et al., 2024). Early works to mitigate hallucinations focused on fine-grained modality alignment (Rohrbach et al., 2018) and reducing co-occurrence biases (Kim et al., 2023) in small-scale models. More recent strategies involve hallucination-related datasets for fine-tuning (Gunjal et al., 2024), post-hoc revisors (Zhou et al., 2024), and adopting RLHF (Yu et al., 2024a). LURE (Zhou et al., 2023) trains a reviser to edit the possible hallucinated words in the responses. While effective, these methods are resource-intensive. Attentional intervention strategies (Zhang et al., 2024a; Xing et al., 2024), represented by OPERA (Huang et al., 2024a), are simpler and do not require additional data or training, but have a higher inference latency. CD-based approaches (Chuang et al., 2023; Chen et al., 2024b; Neo & Chen, 2024), represented by ICD (Wang et al., 2024) and VCD (Leng et al., 2024), adjust the decoding distribution to mitigate hallucinations in MLLM, but this does not consistently improve performance as it introduces potential noise into the CD.

**Comparisons.** Distinct from these methods, our proposed MemVR offers a novel decoding strategy that effectively reduces visual hallucinations (VH) without necessitating extra models, data, and training. Table 2 illustrates the differences and advantages of our method compared to recent representative SOTA approaches. DoLa (Chuang et al., 2023) targets hallucinations in LLM and is negative for alleviating VH, although it exhibits low latency. Attentional intervention methods (Zhang et al., 2024a; Xing et al., 2024), represented by OPERA (Huang et al., 2024a), have achieved considerable progress in mitigating VH, but are limited to the problem of high latency. CD-based approaches, whether modifying the textual input (Wang et al., 2024; Kim et al., 2024) or visual input (Leng et al., 2024; Chen et al., 2024b; Neo & Chen, 2024), both aim to reduce the output probability of

incorrect tokens through the comparison of logits. However, this brings two challenges, one is that the CD strategy may introduce noise to the output distribution, thus losing the original capability, and the other is that the CD-based mechanisms often require multiple rounds of inference to obtain several pairs of logits for contrasting, resulting in a high latency. Significantly, these methods work negatively or fail in general-purpose testing. Compared with them, MemVR stands as "*a paradigm of effectiveness and efficiency*" in visual hallucination mitigation and general improvement.

## 3 Background and Motivation

### 3.1 Problem Formulation

Given an MLLM $\mathcal{M}_\theta^{\mathrm{MLLM}}$ parameterized by $\theta$, with a general architecture consisting of a text embedding layer, a vision encoder, a vision-text interface module, a text decoder consisting of $L$ number of transformer layers, and an affine layer $\varsigma(\cdot)$ which predicts the distribution of the next token. For an image-grounded text generation task, given a textual query $x$ and an input image $v$, MLLM first extracts vision features of $v$ by the vision encoder, and then converts them into visual tokens $z_v$ by MLP or Q-Former (Wadekar et al., 2024) modules. Aligned vision tokens $z_v$ are concatenated with the query $x$ as input to the text decoder, and finally decoded into a textual response $y$ autoregressive, which is formulated as follows:

$$y_t \sim p_\theta(\cdot|v, x, y_{<t}) \propto \mathrm{softmax}(f_\theta(\cdot|v, x, y_{<t})), \quad (1)$$

where $y_t$ indicates the $t^{th}$ token, $y_{<t}$ is the token sequence generated up to time step $t$, and $f_\theta$ is the logit distribution, *i.e.*, unnormalized log-probabilities produced by $\mathcal{M}_\theta^{\mathrm{MLLM}}$.

When the text generation $y$ is inconsistent or in conflict with the input image $v$, MLLM is believed to present hallucination issues. The objective of visual hallucination mitigation is to minimize the appearance of incorrect or conflicting

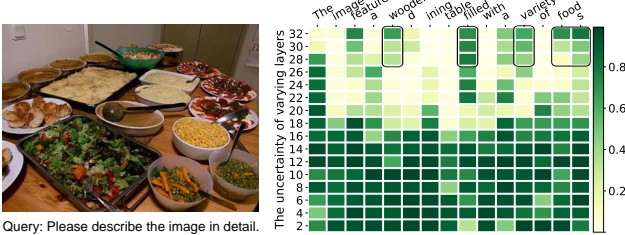

Query: Please describe the image in detail.

Figure 3: Uncertainty of different layers to predict the next token. Rows denote indices of the early layers, and column names are decoded tokens in each step. Uncertainty distribution is dynamic.

tokens, ensure faithfulness to $v$ when answering the query $x$, and simultaneously maintain high-quality text generations.

## 3.2 Why Does MLLM Generate Hallucination?

Hallucinations in MLLM are caused by multiple factors, including inherent biases in the training data (Zhou et al., 2023), visual uncertainty resulting from the model's statistical bias and priors (Leng et al., 2023), and the limitations of current models in accurately discerning context and fact throughout the output generation process (Daunhawer et al., 2021). Upon a more in-depth analysis, we consider that the imbalance of modalities in MLLM and the autoregressive characteristic of language models are likely crucial factors causing their hallucinatory phenomena. Taking image and text as an example, since an image possesses a much higher information density than a piece of text, it is reasonable to suppose that *LLMs struggle to understand or memorize vision information compared to text*. Moreover, autoregressive decoding causes MLLMs to increasingly depend on textual information, including query $x$ and growing tokens $y_{<t}$, inevitably decreasing reliance on visual input. As attention sinks (Zhang et al., 2024b) and PAI (Liu et al., 2024c) proposed to pay more attention to images, both reflect the fact that hallucinations may be caused by modality imbalances.

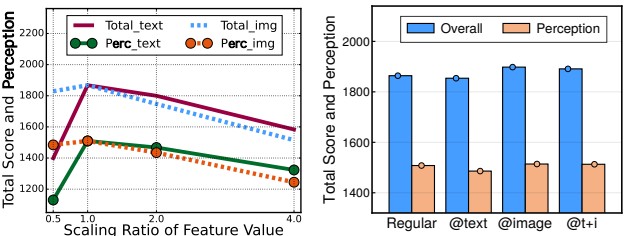

Figure 4: (Left) Performance under different scaling ratios to text / image feature value on MME; (Right) Performance changes when *look-twice* to text / image / text+image (*i.e,* @t+i), respectively.

To further verify this conjecture, we scale the feature values of different modalities up or down to simulate the modality imbalance phenomenon. As shown in the left of Figure 4, it can be observed that: when the image and text features are scaled up proportionally, modality balance is disrupted in both cases, yet the performance decline caused by magnifying the image features is more significant than that of text features; however, when the image and text features is

scaled down proportionally, the contraction on text features led to a drastic performance drop, while the scaling down of visual features had a relatively minor impact. We can obtain three findings in this experiment: ① modality imbalance issues cause hallucinations in MLLMs, ② text decoders (*i.e.*, LLMs) are more text-informed, ③ LLMs are harder to comprehend the visual modality than textual inputs.

This modality imbalance leads to a substantial deviation from the accurate representation of visual input, eventually giving rise to hallucinations, as manifested by the phenomenons in the aforementioned studies (Zhou et al., 2023). This is especially evident in generating longer responses, explaining the correlation between higher VH and larger maximum token lengths, as presented in Huang et al. (2023).

## 3.3 The Pattern of Hallucinations in MLLM

In order to further explore the potential pattern of hallucinations in MLLM, this study employs uncertainty as the metric. As findings of Chen et al. (2024a) in LLMs: *incorrect tokens generally exhibit higher entropy than correct ones*, we also observe this phenomenon in MLLMs, the visualization case is shown in Figure 5. From Figure 5, it can be noticed that when the model generates illusion tokens, *i.e.*, *a, pom* -eg-ran-ate (~~pomegranate~~), the uncertainty in the middle and last layers of the model is high in MLLM.

**Uncertainty quantification.** Following the DoLa (Chuang et al., 2023), we compute the probability of the next token via the vocabulary head $\varsigma(\cdot)$ on each layer during reasoning. Then, we introduce an entropy-based metric (Farquhar et al., 2024) to quantify the output uncertainty of each layer in the text decoder as $u = \sum -p_i \log p_i / \log N$, where $\{p_i\}_{i=1}^N$.

In addition, in the context of tokens involving *objects, attributes or relations*, uncertainty is also high. We conduct a preliminary analysis with 32-layer LLaVA-1.5-7B. Specifically, we compute the uncertainty in the output distributions of early layers. Figure 3 shows the uncertainty scores of different early layers when decoding the answer, we can observe that the computed uncertainty remains relatively high in later layers when predicting key entity objects, attributes, or relations, such as *wooden, filled, foods*. This phenomenon suggests that LLM is still uncertain about its

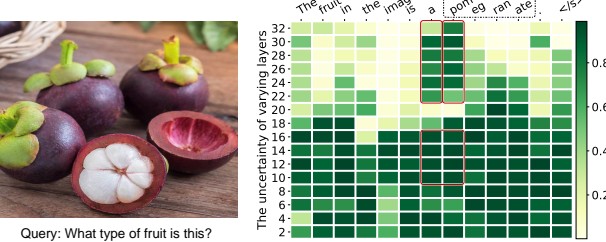

Query: What type of fruit is this?

Figure 5: Uncertainty distribution across layers during token reasoning in hallucinations. Red-outlined regions show higher uncertainty in middle and late layers for hallucinatory tokens: *a, pom*.

predictions in the last few layers and may inject more factual knowledge into the predictions. On the other hand, when predicting function words and those tokens copied from the question, *e.g.*, *image, with*, we observe that the uncertainty becomes very low from the middle layers. This finding implies that the model is deterministic for easy-to-predict tokens at the intermediate layer and keeps the distribution of outputs almost constant at higher layers, however, it is more uncertain for difficult-to-predict key tokens and may constantly change its predictions until the final layer.

### 3.4 Refresh Visual Memory Alleviates Hallucination

Based on the findings, we propose that both overreliance on textual information and schema imbalance contribute to a phenomenon akin to "amnesia" about visual information. It is easy to understand that from the shallow to the deep layers of the Transformer, attention is progressively biased towards textual tokens, resulting in visual tokens at the deeper layer that hardly affect the results, which is consistent with how the attention intervention strategies work. To address this, we try to refresh visual information for trustworthy answers when the model encounters high uncertainty.

We verify this hypothesis through an empirical study. As shown in the right of Figure 4, the proposed *look-twice* strategy is used for combinations of different modalities, including text, image, text+image, where it can be found that the best performance improvement is achieved while only replenishing image information to the model, *i.e.*, refreshing visual memory can effectively alleviate hallucinations in MLLMs. This study lends further credence to our conjecture regarding the causes of hallucinations in Sec. §3.2.

## 4 Methodology

Inspired by the common cognition: *when the memory of an image seen the moment before is forgotten, people will look at it again for factual answers*, we design *look-twice* mechanism, which treats visual tokens as supplementary evidences, re-injecting them into MLLMs through FFN as "key-value memory" at the middle trigger layer. This *look-twice* mechanism occurs when the model exhibits high uncertainty during inference, effectively enhancing factual alignment and modality balance. All details are listed below.

### 4.1 Preliminary: Reformulation of FFN

Vanilla FFN comprises two fully connected layers with non-linear activation in between. We suppose $\boldsymbol{x} \in \mathbb{R}^d$ as an input token of the FFN, and FFN function can be formulated as

$$\text{FFN}(\boldsymbol{x}) = \phi\left(\boldsymbol{x}\boldsymbol{W}_1\right)\boldsymbol{W}_2^{\top}, \quad (2)$$

where $\phi$ is activation function like ReLU or SiLU (Liu et al., 2020), and $\boldsymbol{W}_1, \boldsymbol{W}_2 \in \mathbb{R}^{d \times D}$ are the weight matrices, in

usual $D = 4d$. Peculiarly, $\boldsymbol{W}_1$ and $\boldsymbol{W}_2$ can be rewritten as

$$\boldsymbol{W}_1 = (\boldsymbol{k}_1, \boldsymbol{k}_2, \ldots, \boldsymbol{k}_D), \boldsymbol{W}_2 = (\boldsymbol{v}_1, \boldsymbol{v}_2, \ldots, \boldsymbol{v}_D), \quad (3)$$

where $\boldsymbol{k}_i, \boldsymbol{v}_i \in \mathbb{R}^d$ denote entries of key and value, respectively. As a result, the FFN can be reformulated as

$$\text{FFN}(\boldsymbol{x}) = \sum \phi\left(\langle \boldsymbol{x}, \boldsymbol{k}_i \rangle\right) \cdot \boldsymbol{v}_i . \quad (4)$$

Thus, the FFN function can be construed as using input $\boldsymbol{x}$ as a query to measure similarity with keys, find matching values, and gather values by similarity, which works like a key-value memory storing the factual knowledge as found in previous studies (Geva et al., 2021; Jie et al., 2024).

### 4.2 FFN with Visual Retracing

Motivated by the findings in section §3.2 and §3.4, we propose Visual Retracing (VR), *i.e.*, reinjecting visual evidence into the middle layer of the text decoder during elevated uncertainty during reasoning. This strategy treats visual tokens as anchors to recalibrate off-target predictions and reduces uncertainties in *object, attribute, relationship* tokens. Experimental results also demonstrate that our method reduces uncertainty and alleviates hallucinations as shown in Figure 8. The reason we call this pattern of reinjecting visual evidence "visual retracing" is that the model finds and refreshes key visual memories based on the hidden states in this process. In particular, inspired by the fact that FFN executes analogous retrieval from its key-value memory, we consider VR to serve as a simplified and efficient information re-retrieval process. Given a hidden token $\boldsymbol{x} \in \mathbb{R}^d$ and dimension-aligned vision tokens $\boldsymbol{z}_v$, FFN with visual retracing at $l$-th layer can be written as follows

$$\text{FFN}^{(l)}(\boldsymbol{x} \propto \boldsymbol{z}_v) = \alpha\underline{\Delta} + (1 - \alpha)\,\text{FFN}^{(l)}(\boldsymbol{x}), \quad (5)$$

where $\boldsymbol{z}_v = (\boldsymbol{z}_{v,1}, \ldots, \boldsymbol{z}_{v,N_v}) \in \mathbb{R}^{d \times N_v}$, $x \propto \boldsymbol{z}_v$ denotes execute VR $\underline{\Delta}$ from $\boldsymbol{x}$ to visual features $\boldsymbol{z}_v$, and $\alpha \in [0, 1]$ denotes injection ratio of visual memory through the FFN layer which proportional to image complexity. Specifically, instead of performing retrieval via cross-attention layers as in previous approaches (Li et al., 2022; Alayrac et al., 2022), we consider a simple retrieval process for VR as,

$$\underline{\Delta}(\boldsymbol{z}_v \mid \boldsymbol{x}) = \sum_{i=1}^{N_v} \phi(\langle \boldsymbol{x}, \boldsymbol{z}_{v,i} \rangle) \cdot \boldsymbol{z}_{v,i}. \quad (6)$$

From the perspective of FFN, visual retracing works by treating $\boldsymbol{x}$ as a query, and $\langle \boldsymbol{z}_{v,i} : \boldsymbol{z}_{v,i} \rangle$ as new key-value entries (visual evidence) to supplement vision-related information in the hidden states. In this information re-retrieval process, MemVR does not introduce any parameters that need to be trained. Notably, since the size of key-value memory $D$ in FFN typically far exceeds the number of visual tokens $N_v$ (for instance, $D = 11008$ in LLaMA-7B and $N_v = 256$ for ViT-L/14, $N_v \ll D$), the computation of MemVR is negligible. Thus, VR operation is more efficient than the cross-attention mechanism with quadratic complexity.

### 4.3 Dynamic Triggered Visual Retracing

To magnify the effectiveness of VR, the optimal strategy should be to trigger VR dynamically based on the token, and the selection of the trigger layer should also be dynamically determined. In practice, we consider that the uncertainty of a candidate layer exceeding the threshold $\gamma$ warrants visual retracing. Inspired by the fact that *early exit* patterns (Elbayad et al., 2020; Schuster et al., 2022) have proven effective in directly employing the language heads $\zeta$ to the hidden states of the middle layers, even without a special training process (Kao et al., 2020), we compute the uncertainty of the next token probability on the early layers for reasoning. We utilize layer-specific uncertainty to allow for dynamic premature layer injection at each time step.

**Dynamic Triggered MemVR.** For MLLMs with different numbers of layers, as Algorithm 1 shown, the dynamic triggered MemVR strategy called MemVR-dynamic, identifies desirable premature layers among the candidate layers for visual retracing based on output uncertainty of different layers, thus better amplifying the effect of visual retracing.

---

**Algorithm 1** Dynamic Triggered MemVR Strategy

---

**Require:** MLLM $\mathcal{M}_\theta^{\text{MLLM}}$, text query $x$, image input $v$.
**output** Model response $y_t^b$.
1: At every decoding step $t$:
2: Initial set to *trigger* = TRUE.
3: **for** $l = 1$ to $L - 1$ **do**
4:     $u^{(l)} = \sum -p_\theta^{(l)} \log p_\theta^{(l)} / \log N$.    % uncertainty
5:     **if** *trigger* == TRUE and $u^{(l)} > \gamma$ **then**
6:        Execute $\underline{\Delta}(z_v \mid h_t^{(l+1)})$ {§4, Eq. (6)}
7:        Select $\text{FFN}^{(l+1)}(h_t^{(l+1)} \propto z_v)$ {§4, Eq. (5)}
8:        *trigger* = FALSE    % set to only look-twice
9:     **end if**
10: **end for**
11: $\mathcal{M}_\theta^{\text{MLLM}}$ decoding, obtain current token $y_t^b$.

---

**Static Triggered MemVR.** Besides MemVR-dynamic, another more straightforward strategy worth considering is to perform a brute-force experiment on all possible early layers using a validation set and selecting the layer with the best average performance. We refer to this simple strategy as MemVR-static. Nonetheless, MemVR-static exhibits two notable limitations. Firstly, it demands more comprehensive hyperparameter tuning across different layers. Secondly, the optimal layer is extremely sensitive to the data distribution, which mandates the use of an in-distribution validation set.

**MemVR-dynamic vs. -static.** In contrast to MemVR-static, MemVR-dynamic strategy effectively alleviates these challenges. It achieves this by narrowing down the layer search space and enhancing robustness, all without relying on an in-distribution validation set. Empirical comparisons between

the proposed MemVR implemented with the dynamic and static strategies are presented in Section §5.4 and Table 10.

### 4.4 Theoretical Analysis

To further understand why MemVR effectively mitigates hallucinations and performs robustly on general benchmarks, we explain these phenomena using three theorems below.

**Theorem 4.1.** *Let $x$ be the hidden states of FFN and $\hat{x}$ be after reinjecting visual evidence $z_v$. MemVR enhances Mutual Information (MI) between $\hat{x}$ and $z_v$ as:*

$$I(\hat{x}; z_v) \geq I(x; z_v). \tag{7}$$

**Theorem 4.2.** *Let $y$ be the target output dependent on hidden states. If MI between $x$ and $z_v$ increases, then conditional entropy $H(y \mid x)$ decreases with*

$$H(y \mid \hat{x}) \leq H(y \mid x). \tag{8}$$

**Theorem 4.3.** *Within the Information Bottleneck (IB) framework, the loss of objective function, represented by the notation $\mathcal{L}(T)$, is optimized by MemVR, which is defined as $\mathcal{L}(\hat{x}) \leq \mathcal{L}(x)$, where $\mathcal{L}(x) = I(x; c) - \beta I(x; y)$ is IB loss, $c$ denotes input embedding and $\beta$ is a trade-off parameter.*

**Intuition:** The proofs are provided in Appendix A. The theoretical basis of MemVR draws support from the DPI (Cover et al., 1991) and the contraction properties of stochastic mappings in deep neural networks, as evidenced in numerous IB-related studies (Achille & Soatto, 2018). By enhancing MI and reducing uncertainty in hidden states, MemVR effectively alleviates hallucination while maintaining efficiency.

## 5 Experiments

### 5.1 Experiment Setup

**Datasets and Evaluation.** To rigorously assess the effectiveness of our proposed MemVR, we conduct a comprehensive set of experiments across POPE benchmark (Li et al., 2023b), CHAIR (Rohrbach et al., 2018), VizWiz-VQA (Gurari et al., 2018), MME (Fu et al., 2023), MMBench (Liu et al., 2023b), MM-Vet (Yu et al., 2024b), LLaVA-Bench (in-the-wild) (Liu et al., 2024b), HallusionBench (Guan et al., 2024). More details can be found in Appendix B.1.

**Implementation Details.** Usually, we set $\gamma$=0.75. All settings of baseline methods follow the default configurations from the original papers. More details are in Appendix B.2.

Table 3: CHAIR evaluation results of different methods.

| Methods | CHAIR$_S$ ↓ | CHAIR$_I$ ↓ | Average ↓ | Len | Recall ↑ |
|---------|-------------|-------------|-----------|-----|----------|
| LLaVA-1.5 | 50.0 ↓0.0 | 15.4 ↓0.0 | 32.7 ↓0.0 | 100.6 | 77.1 ↑0.0 |
| + OPERA | 47.8 ↓2.2 | 14.6 ↓0.8 | 31.2 ↓0.5 | 98.6 | 76.8 ↓0.3 |
| + VCD | 48.6 ↓1.4 | 14.9 ↓0.5 | 31.8 ↓0.5 | 100.4 | 77.3 ↑0.2 |
| + ICD | 56.2 ↑6.2 | 16.3 ↑0.9 | 36.3 ↑3.6 | 103.4 | 16.3 ↓60. |
| + MemVR | 46.6 ↓3.4 | 13.0 ↓2.4 | 29.8 ↓0.5 | 99.6 | 80.8 ↑3.7 |

Table 4: Performance evaluation on POPE benchmark. The best results are in green. We report accuracy and f1-score under three settings, e.g., *Random*, *Popular*, *Adversarial*, and also record the *Average*, to show the robustness of MemVR compared with baseline methods.

| Evaluation | Methods | Random | | Popular | | Adversarial | | Average | |
|---|---|---|---|---|---|---|---|---|---|
| | | Accuracy ↑ | F1-score ↑ | Accuracy ↑ | F1-score ↑ | Accuracy ↑ | F1-score ↑ | Accuracy ↑ | F1-score ↑ |
| MSCOCO | LLaVA-1.5-7B | 83.49 ↑0.0 | 82.28 ↑0.0 | 79.98 ↑0.0 | 79.34 ↑0.0 | 76.03 ↑0.0 | 76.26 ↑0.0 | 79.83 ↑0.0 | 79.29 ↑0.0 |
| | OPERA (Huang et al., 2024a) | 87.53 ↑4.0 | 86.45 ↑4.2 | 84.21 ↑4.2 | 83.50 ↑4.2 | 80.88 ↑4.9 | 80.69 ↑4.4 | 84.21 ↑4.4 | 83.55 ↑4.3 |
| | ICD (Wang et al., 2024) | 84.87 ↑1.4 | 83.27 ↑1.0 | 82.93 ↑3.0 | 81.45 ↑2.1 | 81.07 ↑5.0 | 79.96 ↑3.7 | 82.96 ↑3.1 | 81.56 ↑2.3 |
| | VCD (Leng et al., 2024) | 86.84 ↑3.4 | 86.83 ↑4.6 | 82.65 ↑2.7 | 83.37 ↑4.0 | 77.31 ↑1.3 | 79.28 ↑3.0 | 82.27 ↑2.4 | 83.16 ↑3.9 |
| | MemVR (ours) | 88.50 ↑5.0 | 87.34 ↑5.0 | 87.10 ↑7.1 | 86.01 ↑6.7 | 85.20 ↑9.2 | 84.28 ↑8.0 | 86.93 ↑7.1 | 85.88 ↑6.6 |
| A-OKVQA | LLaVA-1.5-7B | 83.45 ↑0.0 | 82.56 ↑0.0 | 79.90 ↑0.0 | 79.59 ↑0.0 | 74.04 ↑0.0 | 75.15 ↑0.0 | 79.13 ↑0.0 | 79.10 ↑0.0 |
| | OPERA (Huang et al., 2024a) | 88.27 ↑4.8 | 87.54 ↑5.0 | 85.17 ↑5.3 | 84.74 ↑5.2 | 79.37 ↑5.3 | 79.97 ↑4.8 | 84.27 ↑5.1 | 84.08 ↑5.0 |
| | ICD (Wang et al., 2024) | 85.57 ↑2.1 | 85.06 ↑2.5 | 81.93 ↑2.0 | 81.95 ↑2.4 | 77.43 ↑3.4 | 78.99 ↑3.8 | 81.64 ↑2.5 | 82.00 ↑2.9 |
| | VCD (Leng et al., 2024) | 86.15 ↑2.7 | 86.34 ↑3.8 | 81.85 ↑2.0 | 82.82 ↑3.2 | 74.97 ↑0.9 | 77.73 ↑2.6 | 80.99 ↑1.9 | 82.30 ↑3.2 |
| | MemVR (ours) | 91.10 ↑7.7 | 90.83 ↑8.3 | 87.33 ↑7.4 | 87.43 ↑7.8 | 80.20 ↑6.2 | 81.66 ↑6.5 | 86.21 ↑7.1 | 86.64 ↑7.5 |
| GQA | LLaVA-1.5-7B | 83.73 ↑0.0 | 82.95 ↑0.0 | 78.17 ↑0.0 | 78.37 ↑0.0 | 75.08 ↑0.0 | 76.06 ↑0.0 | 78.99 ↑0.0 | 79.13 ↑0.0 |
| | OPERA (Huang et al., 2024a) | 88.27 ↑4.5 | 87.52 ↑4.6 | 83.07 ↑4.9 | 82.93 ↑4.6 | 80.77 ↑5.7 | 81.05 ↑5.0 | 84.04 ↑5.1 | 83.83 ↑4.7 |
| | ICD (Wang et al., 2024) | 84.90 ↑1.2 | 84.22 ↑1.3 | 78.37 ↑0.2 | 78.81 ↑0.4 | 75.97 ↑0.9 | 76.93 ↑0.9 | 79.75 ↑0.8 | 79.99 ↑0.9 |
| | VCD (Leng et al., 2024) | 86.65 ↑2.9 | 86.99 ↑4.0 | 80.73 ↑2.6 | 82.24 ↑3.9 | 76.09 ↑1.0 | 78.78 ↑2.7 | 81.16 ↑2.2 | 82.67 ↑3.5 |
| | MemVR (ours) | 89.60 ↑5.9 | 89.32 ↑6.4 | 84.63 ↑6.5 | 84.98 ↑6.6 | 81.53 ↑6.4 | 82.48 ↑6.4 | 85.25 ↑6.3 | 85.59 ↑6.5 |

## 5.2 Results on Hallucination Benchmarks

We conduct hallucination evaluations on CHAIR, POPE, and HallusionBench with results presented in Table 3, Table 4 and Table 5. In the POPE evaluation, MemVR demonstrates robust effects. Its performance significantly exceeds baseline results, with an average accuracy increase of up to +7.0% and an F1-score increase of up to +7.5% on the A-OKVQA dataset under the *Random*, *Popular*, and *Adversarial* settings. MemVR clearly outperforms all compared SOTA methods. As shown in Table 3, compared with vanilla LLaVA-1.5, MemVR achieves 6.8% and 15.6% improvement on CHAIR$_S$ and CHAIR$_I$ metrics. Table 5 shows the HallusionBench evaluation results, where MemVR outperforms vanilla LLaVA-1.5 and other compared methods, achieving the best performance and consistent improvement across different metrics such as $hard$aACC and aACC.

Table 5: HallusionBench evaluation results of different methods.

| Methods | fACC ↑ | qACC ↑ | $easy$A ↑ | $hard$A ↑ | aACC ↑ |
|---|---|---|---|---|---|
| LLaVA-1.5 | 17.9 ↑0.0 | 8.13 ↑0.0 | 36.0 ↑0.0 | 36.7 ↑0.0 | 41.5 ↑0.0 |
| + OPERA | 16.2 ↓1.8 | 5.49 ↓2.6 | 37.6 ↑0.9 | 35.4 ↓1.3 | 41.2 ↓0.3 |
| + ICD | 13.9 ↓4.0 | 8.35 ↑0.2 | 36.9 ↑0.2 | 33.5 ↓3.4 | 38.2 ↓3.3 |
| + VCD | 13.9 ↓4.0 | 11.4 ↑3.3 | 33.0 ↓3.7 | 34.7 ↓3.0 | 41.1 ↓0.4 |
| + MemVR | 17.9 ↑0.0 | 9.01 ↑0.9 | 36.9 ↑0.9 | 37.7 ↑1.0 | 42.5 ↑1.0 |

## 5.3 Results on General-purpose Benchmarks

We evaluate the performance of MemVR on general-purpose benchmarks, *i.e.*, LLaVA-Bench, MM-Vet, MME, MM-Bench, and VizWiz. As shown in Table 6, MemVR consistently outperforms competing models on LLaVA-Bench. Besides, MemVR achieves a significant improvement in overall performance listed in Table 7, with an average increase of 6.1% in OCR and spatial awareness tasks, demonstrating superior generalization capabilities. For MME subset evaluations (covering both object-level and attribute-level hallucinations), the results in Figure 2 and Table 8 indicate that MemVR uniformly improves in handling object-level and attribute-level hallucinations, as well as commonsense

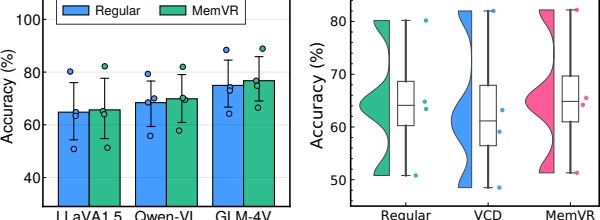

Figure 6: (Left) Comparisons of MemVR with regular LLaVA-1.5, Qwen-VL, and GLM-4V on LLaVA-Bench (in-the-wild). (Right) The comparison between MemVR and VCD on LLaVA-Bench shows that MemVR improves significantly while VCD decreases.

reasoning. The *Existence*, *Count*, and *Color* scores all show significant improvements, where the total score of MME in LLaVA and Qwen-VL increases by 32 and 36, respectively. Importantly, although those comparison methods alleviate some hallucinations, they deteriorate generalizability, *i.e.*, they lead to performance degradation on the general benchmarks. These results indicate that, compared to CD-based methods, MemVR excels at hallucination mitigation and delivers strong performance on general-purpose benchmarks. More experimental results are provided in Appendix B.

Table 6: LLaVA-Bench (In-the-Wild) evaluation results.

| Methods | Convs ↑ | Detail ↑ | Complex ↑ | All ↑ | Average ↑ |
|---|---|---|---|---|---|
| LLaVA-1.5 | 58.8 ↑0.0 | 52.1 ↑0.0 | 74.6 ↑0.0 | 63.4 ↑0.0 | 64.8 ↑0.0 |
| + OPERA | 59.5 ↑0.7 | 49.6 ↓2.5 | 78.6 ↑4.0 | 59.8 ↓3.6 | 64.3 ↓0.5 |
| + ICD | 40.3 ↓18. | 42.2 ↓9.9 | 60.3 ↓14. | 49.8 ↓13. | 56.9 ↓7.9 |
| + VCD | 57.8 ↓1.0 | 50.8 ↓1.3 | 77.9 ↑3.3 | 59.1 ↓4.3 | 63.2 ↓1.6 |
| + MemVR | 63.8 ↑5.0 | 52.6 ↑0.5 | 77.9 ↑3.3 | 64.0 ↑0.6 | 65.2 ↑0.4 |

Table 7: MM-Vet evaluation results, where G denotes language Generation, K: Knowledge S: Spatial awareness, R: Recognition.

| Methods | R ↑ | OCR_S ↑ | OCR_K_R ↑ | OCR_G_S ↑ | Total ↑ |
|---|---|---|---|---|---|
| LLaVA-1.5 | 67.6 ↑0.0 | 17.7 ↑0.0 | 21.2 ↑0.0 | 10.0 ↑0.0 | 31.1 ↑0.0 |
| + OPERA | 61.9 ↓5.7 | 21.5 ↑3.8 | 11.2 ↓10. | 30.0 ↑20. | 32.0 ↑0.9 |
| + ICD | 59.5 ↓8.1 | 17.7 ↑0.0 | 8.8 ↓12.4 | 40.0 ↑30. | 25.9 ↓5.2 |
| + VCD | 62.2 ↓5.4 | 15.8 ↓1.9 | 17.5 ↓3.7 | 60.0 ↑50. | 30.2 ↓1.1 |
| + MemVR | 70.3 ↑2.7 | 23.8 ↑6.1 | 21.2 ↑0.0 | 30.0 ↑20. | 32.4 ↑1.3 |

Table 8: Performance evaluation on MME Hallucination subset, MM-Vet, Vizwiz, and MMBench. The best results are in green.

| Evaluation | Methods | MME-Hall | Object-Level | | Attribute-Level | | MM-Vet | Vizwiz | MMBench |
|---|---|---|---|---|---|---|---|---|---|
| | | Total ↑ | Existence ↑ | Count ↑ | Position ↑ | Color ↑ | Accuracy ↑ | Accuracy ↑ | Accuracy ↑ |
| LLaVA-1.5 | Regular | 643.3 ↑0.0 | 190.0 ↑0.0 | 155.0 ↑0.0 | 128.3 ↑0.0 | 170.0 ↑0.0 | 31.10 ↑0.0 | 50.00 ↑0.0 | 62.80 ↑0.0 |
| | OPERA | 610.0 ↓33. | 195.0 ↑5.0 | 128.3 ↓26. | 121.7 ↓6.6 | 165.0 ↓5.0 | 32.00 ↑0.9 | 50.76 ↑0.8 | 62.80 ↑0.0 |
| | ICD | 583.3 ↓60. | 185.0 ↓5.0 | 130.0 ↓25. | 121.7 ↓6.6 | 146.7 ↓23. | 25.90 ↓5.2 | 37.62 ↓12. | 39.78 ↓23. |
| | VCD | 648.3 ↑5.0 | 190.0 ↑0.0 | 155.0 ↑0.0 | 133.3 ↑5.0 | 170.0 ↑0.0 | 30.20 ↓0.9 | 44.90 ↓5.1 | 54.21 ↓8.6 |
| | MemVR (ours) | 648.3 ↑5.0 | 190.0 ↑0.0 | 155.0 ↑0.0 | 133.3 ↑5.0 | 170.0 ↑0.0 | 32.40 ↑1.3 | 51.50 ↑1.5 | 63.75 ↑0.9 |
| Qwen-VL | Regular | 618.3 ↑0.0 | 175.0 ↑0.0 | 140.0 ↑0.0 | 123.3 ↑0.0 | 180.0 ↑0.0 | 49.00 ↑0.0 | 66.05 ↑0.0 | 56.53 ↑0.0 |
| | OPERA | - ↑0.0 | - ↑0.0 | - ↑0.0 | - ↑0.0 | - ↑0.0 | - ↑0.0 | - ↑0.0 | - ↑0.0 |
| | ICD | 616.7 ↓1.7 | 170.0 ↓5.0 | 138.3 ↓1.7 | 148.3 ↑25. | 160.0 ↓20. | 31.70 ↓17. | 29.37 ↓36. | 13.32 ↓43. |
| | VCD | 648.3 ↑30. | 175.0 ↑0.0 | 130.0 ↓10. | 153.3 ↑30. | 190.0 ↑10. | 34.60 ↓14. | 34.54 ↓31. | 39.18 ↓17. |
| | MemVR (ours) | 638.3 ↑20. | 185.0 ↑10. | 145.0 ↑5.0 | 123.3 ↑0.0 | 185.0 ↑5.0 | 49.60 ↑0.6 | 66.36 ↑0.3 | 56.44 ↓0.1 |

## 5.4 Ablation Studies

To validate the effectiveness across the factors we introduced in our MemVR, we conducted in-depth ablation experiments as detailed in Figure 7 and Table 10. All ablation experiments are conducted on LLaVA-1.5-7B, and the performance is assessed across benchmarks.

**Impact of Threshold and Injection Ratio**. Fig. 7 shows the performance under different injection ratios $\alpha$ and thresholds $\gamma$, we find that setting $\gamma$ between 0.6 and 0.95 can improve performance, with the optimal threshold around 0.75. Understandably, a high threshold makes VR hard to trigger, while a low threshold would trigger VR at an earlier layer. For injection rate $\alpha$, we find a positive effect of between 5% and 35%, and a negative effect above 35%, which suggests that there is an upper limit to the supplementation of visual memory. To drop the hyperparameter $\alpha$ and derive the dynamic injection ratio, we calculate $\alpha$ by $\alpha = 2(u - \gamma)$, where $u$ denotes uncertain score and this variant named MemVR$^\dagger$, and the results are present in Table 9.

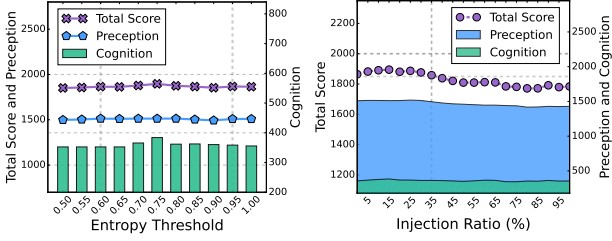

Figure 7: (Left) Results under different thresholds $\gamma$. (Right) Results under different injection ratios $\alpha$, both on MME benchmark.

Table 9: MemVR$^\dagger$ performance on different benchmarks.

| Methods | MME (Overall) | POPE (COCO) | Hallusion -Bench | LLaVA -Bench | POPE (GQA) |
|---|---|---|---|---|---|
| LLaVA-1.5 | 1864.6 | 83.49 | 41.45 | 64.80 | 83.73 |
| + MemVR | 1896.7 | 88.50 | 42.34 | 65.17 | 89.60 |
| + MemVR$^\dagger$ | 1894.2 | 88.40 | 42.27 | 65.87 | 89.57 |

**Impact of VR Strategy**. MemVR triggers VR dynamically using layer entropy. Tab. 10 shows that dynamic VR out-

performs layer-fixed strategy, achieving the highest total score on MME benchmark, where Static-# denotes executing VR on the # layer and Static-$\phi$ means running VR on the specific layer with best performance. This indicates that dynamic VR leveraging layer entropy offers a more effective mechanism, adapting better to different scenarios and achieving optimal performance compared to static VR.

Table 10: MemVR performance with different VR strategies.

| MME | Static-7 | Static-15 | Static-23 | Static-$\phi$ | Dynamic |
|---|---|---|---|---|---|
| Cognition | 347.1 | 352.14 | 357.9 | 362.9 | 383.9 |
| Perception | 1500.5 | 1529.1 | 1500.3 | 1526.4 | 1512.8 |
| Total score | 1847.6 | 1881.2 | 1858.1 | 1889.2 | 1896.7 |

## 5.5 Inference Latency

MemVR operates dynamically based on uncertainty. It uses VR when layer uncertainty exceeds threshold $\gamma$; if uncertainty stays low across all layers, indicating high model confidence, it's not triggered. This mechanism enables efficient inference without extra computation. Different from CD-based and Att-intervention paradigms, which need multi-round inferences or have rollback-induced exponential overheads, our proposed *look-twice* mechanism only requires one regular inference. The comparisons on latency, throughput, time cost, and memory are shown in Table 1.

## 5.6 Cases Study

**MemVR reduces uncertainty during inference.** As shown in Figure 8, MemVR effectively reduces uncertainty after the 8th layer triggered VR, further confirming our analysis about the pattern of hallucinations in Section §3.3.

**Long-text capability.** Beyond single-word QA benchmark evaluation, we explore models' capacity for comprehensive long-text generation in various tasks. As shown in Figure 9, MemVR can accurately identify image details relevant to questions. In contrast, as detailed in Appendix B, Qwen-VL-Chat struggles to generate detailed image descriptions in VCD, especially when nuanced image interpretation is needed. This indicates MemVR has better cross-architecture adaptability and more reliable long-text generation ability.

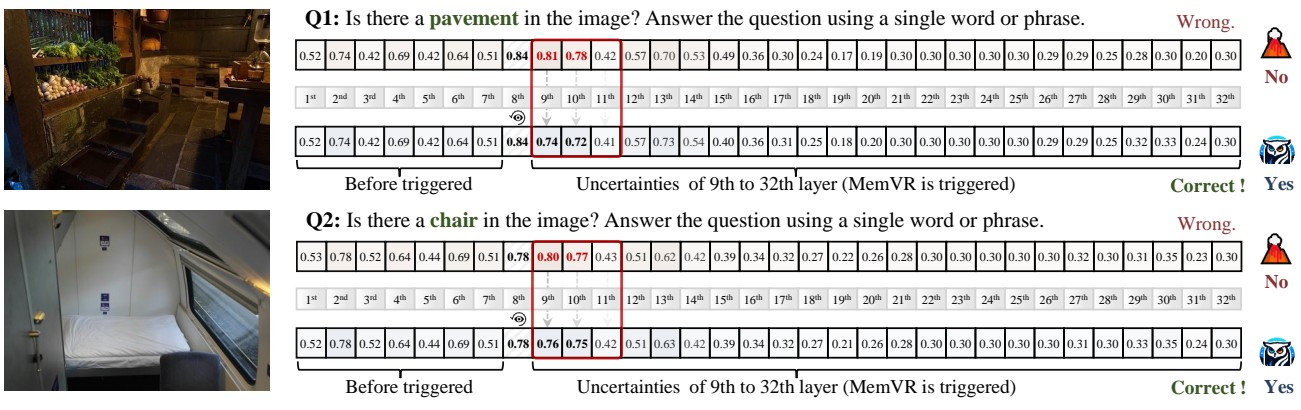

Figure 8: Visualisation of uncertainty across layers without and with MemVR. MemVR effectively reduces uncertainty after 8th layer.

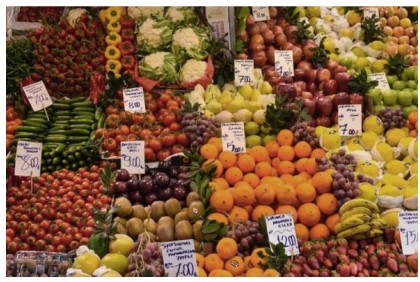

Question: What is the price for tomatoes?

**Ground Truth:** 8.00

**GLM-4v-9B Default:** The price for tomatoes is 9.00, as indicated by the white sign with black lettering among the red tomatoes.

**GLM-4v-9B + VCD:** The price for tomatoes is not clearly visible in the image provided. There are price tags for some of the other fruits and vegetables, but the tomatoes are either not labeled or the label is not in focus. If you need the exact price for tomatoes, I would recommend visiting the market in person or checking with the vendor.

**GLM-4v-9B + MemVR (Ours):** The price for tomatoes is 8.00, as indicated by the white sign with black lettering among the red tomatoes.

Figure 9: A case study in long text generation. Compared with VCD, MemVR could effectively mitigate hallucinations.

**Analysis of failure cases.** We have collected failure cases from the MME benchmark, in the 'Celebrity,' 'Scene,' and 'Landmark' sub-tasks, where MemVR underperforms compared to the default model. We categorize MemVR's failure cases into two types. Type 1 occurs when the default model is right but MemVR is wrong, due to over-disturbing the default model's reasoning. In these instances, the original visual features are sufficient for reasoning, and the reinjected tokens inadvertently disrupt this process, leading to errors. We're exploring ways to reduce such disturbances. For failure type 2, it happens when both are wrong, caused by image complexity or MLLM's original knowledge gaps.

### 5.7 Limitations and Further Discussions

While MemVR shows substantial potential, it is not devoid of limitations. A primary challenge resides in the complex task of identifying the optimal hyperparameters. These hyperparameters include the injection ratio of visual information, and the strategy for selecting the triggered layers. This represents the focal point of our subsequent research efforts, as we work to perfect MemVR.

Additionally, although our research focused on MLLMs with visual inputs, theoretically, MemVR can be expanded to more modalities, *e.g.*, *listen-twice* for audio, *scan-twice* for spatial perception, *check-twice* for fMRI, etc. This opens up an exciting avenue for future work, where we plan to

extend MemVR's framework to these diverse input formats and assess its efficacy across a broader range of tasks.

## 6 Conclusion

This paper proposes a novel decoding paradigm to mitigate hallucination, named MemVR, and present static and dynamic VR strategies that shift hidden states of the intermediate layer in MLLM for self-enhancement, rather than modulating logits directly with CD manner, thus avoiding multi-round decoding. Our experiments, conducted on eight benchmarks, demonstrate the effectiveness of MemVR in mitigating hallucination and improving general performance. Importantly, MemVR is a plug-and-play and task-agnostic method compatible with any MLLM, without extra fine-tuning, emphasizing its widespread applicability.

## Acknowledgement

This work is sponsored by CAAI-Ant Group Research Fund; Guangdong Provincial Department of Education Project (Grant No.2024KQNCX028); Scientific Research Projects for the Higher-educational Institutions (Grant No.2024312096), Education Bureau of Guangzhou Municipality; Guangzhou-HKUST(GZ) Joint Funding Program (Grant No.2025A03J3957), Education Bureau of Guangzhou Municipality; Fundamental Research Funds for National Universities, CUG (Grant No.2024XLB7).

## Impact Statement

This paper proposes MemVR to alleviate hallucinations in MLLMs. The work has potential wider implications. Ethically, it may inherit biases from pre-trained models such as CLIP, risking unfair representation. There's also concern that it could be misused to generate disinformation. In terms of societal impact, MemVR can improve system reliability in safety-critical areas such as healthcare.

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

# A    Theoretical Analysis of MemVR

In Multimodal Large Language Models, hallucinations often arise due to insufficient alignment between visual inputs and the model's internal representations. This paper provides a rigorous theoretical analysis demonstrating that re-injecting visual features into the intermediate layers of MLLMs mitigates hallucinations and enhances representation capability.

We demonstrate that MemVR increases Mutual Information (MI) between the hidden states and visual tokens, decreasing the conditional entropy of outputs given the hidden state for fidelity to the visual input. We begin by defining the relevant variables and information-theoretic concepts that will be used throughout the proof as, $X_{vq}$ denote concatenated tokens of text and vision, with probability distribution $p(X_{vq})$; $Z_v$ means visual (image) features, with probability distribution $p(Z_v)$; The output hidden states of the Transformer model at layer $k$, defined recursively as: $H_{vq}^{(k)} = f^{(k)}(H_{vq}^{(k-1)}, \mathbf{1}_{k=m} Z_v)$, where $\mathbf{1}_{k=m}$ is the indicator function that equals 1 when $k = m$ (the layer where $Z_v$ is rejected) and 0 otherwise, and $Y$ denotes the target output of MLLMs.

The probability of hallucination can be expressed as:

$$P_{\text{hallucination}} = P(Y \neq Y^* \mid X_{vq}),$$

where $Y^*$ is the ground truth output. According to information theory, a higher conditional entropy $H(Y \mid X_{vq})$ indicates greater uncertainty of $Y$ given $X_{vq}$, which increases the probability of hallucination.

**Information Flow of Visual Features.** In a standard Transformer model, the initial input $X_{vq}$ undergoes multiple layers of processing. As the number of layers increases, the initial visual information may gradually diminish (*information forgetting*). In the absence of MemVR, the MI between the hidden states and the visual features $Z_v$ tends to decrease with depth:

$$I(H_{vq}^{(l)}; Z_v) \leq I(H_{vq}^{(l-1)}; Z_v),$$

for $l > 1$. This inequality indicates that in deeper layers, $H_{vq}^{(l)}$ contains less vision-related information.

**Theorem A.1.** *Assume that each Transformer layer acts as a deterministic or stochastic mapping with the Markov property. Then, the mutual information between the hidden states and the visual features decreases with depth:*

$$I(H_{vq}^{(l)}; Z_v) \leq I(H_{vq}^{(l-1)}; Z_v).$$

*Proof.* Each Transformer layer can be modeled as a stochastic mapping (Markov kernel) that processes the input hidden states. Specifically, $H_{vq}^{(l)}$ is a function of $H_{vq}^{(l-1)}$, possibly incorporating additional inputs such as $Z_v$ at specific layers.

According to the Data Processing Inequality (DPI) (Cover et al., 1991), if $A \rightarrow B \rightarrow C$ forms a Markov chain, then:

$$I(A; C) \leq I(A; B).$$

In this context, consider $A = Z_v$, $B = H_{vq}^{(l-1)}$, and $C = H_{vq}^{(l)}$. Since $H_{vq}^{(l)}$ is generated from $H_{vq}^{(l-1)}$ without direct access to $Z_v$, we have the Markov chain $Z_v \rightarrow H_{vq}^{(l-1)} \rightarrow H_{vq}^{(l)}$. Applying DPI yields:

$$I(Z_v; H_{vq}^{(l)}) \leq I(Z_v; H_{vq}^{(l-1)}).$$

Thus, mutual information between the hidden states and the visual features does not increase with depth. $\square$

**Visual Retracing in MLLMs.** We reinject vision tokens $Z_v$ on $l$-th layer ($ahead\_layer \leq l < L$):

$$\hat{H}_{vq}^{(l)} = \text{FFN}^{(l)}(H_{vq}^{(l)} \propto Z_v).$$

MemVR ensures that after the $l$-th layer, $\hat{H}_{vq}^{(l)}$ again contains question-aligned visual information.

**Theorem A.2.** *Let $H_{vq}$ be the hidden states of FFN and $\hat{H}_{vq}$ be after reinjection of visual evidence $Z_v$. MemVR enhances Mutual Information (MI) between $\hat{H}_{vq}$ and $Z_v$: $I(\hat{H}_{vq}; Z_v) \geq I(H_{vq}; Z_v)$.*

*Proof.* We aim to show that reinjecting $Z_v$ at layer $l$ increases the mutual information between the hidden states and $Z_v$ conditioned on $X_{vq}$.

By the definition of conditional mutual information:

$$I(\hat{H}_{vq}^{(l)}; Z_v \mid X_{vq}) = \mathbb{E}_{X_{vq}}[I(\hat{H}_{vq}^{(l)}; Z_v \mid X_{vq} = x)].$$

Similarly,

$$I(H_{vq}^{(l)}; Z_v \mid X_{vq}) = \mathbb{E}_{X_{vq}}[I(H_{vq}^{(l)}; Z_v \mid X_{vq} = x)].$$

Given $\hat{H}_{vq}^{(l)} = \text{FFN}^{(l)}(H_{vq}^{(l)} \propto Z_v)$ denotes the hidden states after utilizing MemVR on $l$-th, reinjection of $Z_v$ introduces a direct dependency between $\hat{H}_{vq}^{(l)}$ and $Z_v$ beyond what is present in $H_{vq}^{(l)}$. Since $\text{FFN}^{(l)}$ is a deterministic function that incorporates $Z_v$, the mutual information $I(\hat{H}_{vq}^{(l)}; Z_v \mid X_{vq})$ is at least as large as $I(H_{vq}^{(l)}; Z_v \mid X_{vq})$. $\hat{H}_{vq}^{(l)}$ retains all information in $H_{vq}^{(l)}$ and additionally incorporates information from $Z_v$. Thus, MemVR ensures that:

$$I(\hat{H}_{vq}^{(l)}; Z_v \mid X_{vq}) \geq I(H_{vq}^{(l)}; Z_v \mid X_{vq}).$$

By directly incorporating $Z_v$ into the computation of $\hat{H}_{vq}^{(m)}$, MemVR ensures that the hidden states retain more information about the visual features relative to the original hidden states $H_{vq}^{(m)}$, thereby increasing $I(\hat{H}_{vq}^{(m)}; Z_v \mid X_{vq})$, enhancing the representation capability and utilizing visual information. $\square$

**Theorem A.3.** *Let $Y$ be the target output dependent on hidden states. If MI between $H_{vq}^{(l)}$ and $Z_v$ increases, then conditional entropy $H(Y \mid H_{vq}^{(l)})$ decreases, leading to a lower probability of hallucinations:*

$$H(Y \mid \hat{H}_{vq}^{(l)}) \leq H(Y \mid H_{vq}^{(l)}).$$

*Proof.* We aim to show that an increase in mutual information between $\hat{H}_{vq}^{(l)}$ and $Z_v$ conditioned on $X_{vq}$ leads to a decrease in the conditional entropy $H(Y \mid \hat{H}_{vq}^{(l)})$. According to the definition of conditional entropy, we have,

$$H(Y \mid \hat{H}_{vq}^{(l)}) = H(Y) - I(Y; \hat{H}_{vq}^{(l)}),$$
$$H(Y \mid H_{vq}^{(l)}) = H(Y) - I(Y; H_{vq}^{(l)}).$$

From Theorem A.2: $\hat{H}_{vq}^{(l)}$ contains more information about $Z_v$, i.e., $I(\hat{H}_{vq}^{(l)}; Z_v) \geq I(H_{vq}^{(l)}; Z_v)$. There is $I(Y; \hat{H}_{vq}^{(l)}) \propto I(\hat{H}_{vq}^{(l)}; Z_v)$, thus we have $I(\hat{H}_{vq}^{(l)}; Y) \geq I(H_{vq}^{(l)}; Y)$. Then, we assume a dependency between $Z_v$ and $Y$, i.e., $I(Z_v; Y) > 0$, and subtract the inequalities, have:

$$\begin{aligned} H(Y \mid \hat{H}_{vq}^{(l)}) &= H(Y) - I(Y; \hat{H}_{vq}^{(l)}) \\ &\leq H(Y) - I(Y; H_{vq}^{(l)}) \\ &= H(Y \mid H_{vq}^{(l)}). \end{aligned}$$

Thus, MemVR reduces the conditional uncertainty of the target output given the intermediate embedding, thereby mitigating the probability of hallucinations and improving the model's predictive capability. $\square$

**Theorem A.4.** *Within the Information Bottleneck (IB) framework, reinjecting $Z_v$ at layer $m$ optimizes the objective function:*

$$\mathcal{L}(\hat{H}_{vq}^{(m)}) \leq \mathcal{L}(H_{vq}^{(m)}),$$

*where the IB objective is defined as:*

$$\mathcal{L}(H) = I(H; X_{vq}) - \beta I(H; Y),$$

*and $\beta$ is a trade-off parameter.*

*Proof.* The Information Bottleneck (IB) objective aims to find a representation $H$ that maximizes the mutual information with the target $Y$ while minimizing the mutual information with the input $X_{vq}$. The optimization objectives before & after MemVR are as follows:

$$\mathcal{L} = I(H_{vq}^{(l)}; X_{vq}) - \beta I(H_{vq}^{(l)}; Y),$$
$$\mathcal{L}_{\diamond} = I(\hat{H}_{vq}^{(l)}; X_{vq}, Z_v) - \beta I(\hat{H}_{vq}^{(l)}; Y),$$

where $I(\hat{H}_{vq}^{(l)}; X_{vq}, Z_v) = I(\hat{H}_{vq}^{(l)}; X_{vq}) + I(\hat{H}_{vq}^{(l)}; Z_v \mid X_{vq})$. The gap in the objective function is:

$$\Delta\mathcal{L} = \mathcal{L}_{\diamond}^{(l)} - \mathcal{L}^{(l)}$$
$$= [I(\hat{H}_{vq}^{(l)}; X_{vq}) + I(\hat{H}_{vq}^{(l)}; Z_v \mid X_{vq}) - \beta I(\hat{H}_{vq}^{(l)}; Y)] - [I(H_{vq}^{(l)}; X_{vq}) - \beta I(H_{vq}^{(l)}; Y)]$$
$$= [I(\hat{H}_{vq}^{(l)}; X_{vq}) - I(H_{vq}^{(l)}; X_{vq})] + I(\hat{H}_{vq}^{(l)}; Z_v \mid X_{vq}) - \beta[I(\hat{H}_{vq}^{(l)}; Y) - I(H_{vq}^{(l)}; Y)].$$

To ensure that $\mathcal{L}_{\diamond}^{(m)} \leq \mathcal{L}^{(m)}$, we require: $\Delta\mathcal{L} \leq 0$. We define the changes in mutual information. Let $\Delta I_X = I(H_{vq}^{(l)}; X_{vq}) - I(H_{vq}^{(l-1)}; X_{vq})$, $\Delta I_Y = I(H_{vq}^{(l)}; Y) - I(H_{vq}^{(l-1)}; Y)$. Note that $I(H_{vq}^{(l)}; Z_v \mid X_{vq}) \geq 0$. For $\Delta I_X$, the change in mutual information between $H_{vq}^{(l)}$ and $X_{vq}$ depends on how much additional information from $Z_v$ affects the dependence on $X_{vq}$. We denote the maximum possible increase as $\Delta I_X^{\max}$. For $\Delta I_Y$, From Theorem A.2, $\Delta I_Y \geq 0$, and suppose we can establish a minimum increase $\Delta I_Y^{\min} > 0$. $I(H_{vq}^{(l)}; Z_v \mid X_{vq})$ represents supplement information about $Z_v$ in $H_{vq}^{(l)}$ that is not already explained by $X_{vq}$, and we denote this maximum as $I_{\max}^{Z|X}$.

To satisfy this inequality, choose $\beta$ such that:

$$\Delta\mathcal{L} \leq 0 \Rightarrow \beta\Delta I_Y \geq \Delta I_X + I(H_{vq}^{(l)}; Z_v \mid X_{vq}). \tag{9}$$

Upper Bound on $\Delta I_X$ and $I(H_{vq}^{(l)}; Z_v \mid X_{vq})$ as $\Delta I_X \leq \Delta I_X^{\max}$, $I(H_{vq}^{(l)}; Z_v \mid X_{vq}) \leq I_{\max}^{Z|X}$. Lower Bound on $\Delta I_Y$ as: $\Delta I_Y \geq \Delta I_Y^{\min} > 0$. Then, we derive the condition with error bounds, for $\Delta\mathcal{L} \leq 0$, it suffices that:

$$\beta\Delta I_Y^{\min} \geq \Delta I_X^{\max} + I_{\max}^{Z|X} \Rightarrow \beta \geq \frac{\Delta I_X^{\max} + I_{\max}^{Z|X}}{\Delta I_Y^{\min}}. \tag{10}$$

This condition provides a lower bound for $\beta$ to ensure that reinjecting $Z_v$ at layer $m$ decreases the IB objective function. By adhering to this condition, MemVR optimizes the IB objective, balancing the trade-off between the compression of input information and the preservation of relevant information for prediction. $\square$

By reducing the IB objective function, the model focuses more on information relevant to predicting $Y$ while compressing irrelevant information. The enhanced mutual information with $Y$ reduces the likelihood of generating hallucinated outputs not supported by the visual input.

**Error Bounds Provide Guarantees.** The upper and lower bounds on mutual information changes ensure that, under specific conditions (e.g., the selection of $\beta$), theoretical improvement holds.

**Estimating the Bounds.**

❶ $\Delta I_Y^{\min}$ requires knowledge of how much additional information about $Y$ is gained by reinjecting $Z_v$. It can be estimated based on the mutual information $I(Z_v; Y)$ and the effectiveness of $H_{vq}^{(m)}$ in capturing information relevant to $Y$.

❷ $\Delta I_X^{\max}$ can be bounded based on the capacity of $H_{vq}^{(m)}$ to represent $X_{vq}$. Specifically, it relates to how much additional information $H_{vq}^{(m)}$ can encode about $X_{vq}$ beyond what was already captured in $H_{vq}^{(m-1)}$.

❸ $H(Z_v)$ is bounded by the entropy of the visual features, as mutual information cannot exceed the entropy of $Z_v$.

Through detailed mathematical derivations and the inclusion of upper and lower error bounds, we have established that:

(a) **Increased Mutual Information:** Reinjecting visual features at an intermediate layer increases the mutual information between the model's embeddings and the visual input.

(b) **Reduced Conditional Entropy:** MemVR reduces the conditional uncertainty of the target output given the intermediate embedding, enhancing the model's predictive accuracy and mitigating hallucination phenomena caused by the forgetting of visual information.

(c) **Optimization within IB Framework:** Within the Information Bottleneck framework, MemVR optimizes the objective function, provided certain conditions on the mutual information changes are met and appropriate choices of the trade-off parameter $\beta$ are made.

## B  Additional Experiments, Results, and Discussions

### B.1  Benchmarks and Metrics

**Datasets.** To rigorously assess the effectiveness of our proposed method, we conduct a comprehensive set of experiments across two benchmarks specifically designed to evaluate hallucination mitigation and five general-purpose benchmarks to gauge the general performance: ① Hallucination benchmarks: Polling-based Object Probing Evaluation (POPE) (Li et al., 2023b), and Caption Hallucination Assessment with Image Relevance (CHAIR) (Rohrbach et al., 2018), Hallusion-Bench (Guan et al., 2024); ② General-purpose benchmarks: VizWiz-VQA (Gurari et al., 2018), MLLM Comprehensive Evaluation (MME) (Fu et al., 2023), Multimodal Benchmark (MMBench) (Liu et al., 2023b), Multimodal Veterinarian (MM-Vet) (Yu et al., 2024b), LLaVA-Bench (in-the-wild) (Liu et al., 2024b).

In this appendix, we provide additional details on the benchmarks referenced in the main paper. To evaluate hallucinations, we employ the following five benchmarks:

**CHAIR** (Rohrbach et al., 2018) evaluates how well the generated captions align with the content of the given image. CHAIR consists of two versions: CHAIRs, which measures the inaccuracies at the sentence level, and CHAIRi, which evaluates at the object level within the sentence by comparing the number of false objects to the total number of objects. For evaluation, we use the val2014 split of the MSCOCO (Lin et al., 2014) dataset, which includes annotations for 80 object categories. We randomly select 500 images from the entire dataset and used the prompt "Please describe this image in detail." for the MLLM. The CHAIR metric includes per-instance evaluation (CHAIRi) and per-sentence evaluation (CHAIRs), defined as follows:

$$\text{CHAIR}_i = \frac{|\{\text{hallucinated objects}\}|}{|\{\text{all objects mentioned}\}|}$$

$$\text{CHAIR}_s = \frac{|\{\text{sentences with hallucinated object}\}|}{|\{\text{ all sentences}\}|}$$

**Polling based Object Probing Evaluation (POPE)** (Li et al., 2023b) is a VQA-based metric proposed to assess hallucinations in MLLMs. This metric evaluates the MLLM's response to the prompt "Is [object] is in this image?" To emphasize that this is a binary VQA task, we appended the prompt with "Please answer yes or no." To select objects referenced in the question prompt, we followed three different sampling options: random, popular, and adversarial. We evaluated performance across all sampling options.

**MLLM Evaluation (MME)** (Fu et al., 2023) evaluates the capabilities of MLLMs, dividing the evaluation into two major categories: perception and cognition. The perception category includes fine-grained tasks such as existence, count, location, rough color, poster, celebrity, scene, landmark, artwork identification, and OCR. The cognition category includes tasks like commonsense reasoning, numerical calculations, text translation, and code reasoning. All questions in this benchmark are structured to be answered with a simple yes or no.

Using the **LLaVA-Bench** (Liu et al., 2024b), we further demonstrated how well our proposed method maintains the language model performance. This benchmark involves posing various situational questions, such as dialogue, detailed descriptions, and complex reasoning, to randomly selected images from the MSCOCO val2014 dataset. A total of 60 questions are used to assess whether the model faithfully follows the instructions. The generated answers are evaluated by comparing them to the responses of a text-only GPT-4 model.

**Candidate Layers**. In dynamic premature layer selection, we partition transformer layers into buckets and select one bucket as the candidate layer set. For 32-layer LLaVA-1.5-7B, we use two buckets:[0,15),[15,31). This design limits the

hyperparameter search space to only 2-4 validation runs. For efficiency, we use a validation set (MME) to select the best bucket.

## B.2 Backbones and Baselines

To evaluate our method, we utilize well-known MLLMs: LLaVA-1.5 (Liu et al., 2024b), Qwen-VL (Bai et al., 2023), and GLM4V (Wang et al., 2023), LLaVA-Next (Li et al., 2024). Further, We compare our methods with classic training-tree SOTA methods designed to mitigate object hallucination, including visual contrastive decoding SOTA VCD (Leng et al., 2024), OPERA (Huang et al., 2024a) based on overconfidence penalty and hindsight allocation. As Dola (Chuang et al., 2023) is layer-wise contrastive decoding for LLMs and performs poorly in MLLMs, it will not be shown in the experiment. Experimental results are obtained and benchmarked using unified implementation.

Greedy search is used as the default decoding strategy in MemVR for all benchmarks. For benchmarks, annotation questions are adapted to MLLM templates. For POPE, COCO, A-OKVQA, and GQA are used, while MMBench_DEV_EN is used for MMBench. MM-Vet is assessed using MM-Vet Online Evaluator, and `gpt4-1106-preview` is used for LLaVA-Bench. CHAIR uses images from COCO Val2014 with the query "Please describe this image in detail". In MemVR, do_sample=False, temperature=0, threshold=0.75, beam=1. All settings of the compared method follow the default configurations from the original papers.

## B.3 Reproducibility

**Implementation details**. We employed greedy search as the default decoding strategy across all benchmark evaluations. For the hallucination benchmarks (POPE, CHAIR and HallusionBench) and general-purpose benchmarks (MME, VizWiz-VQA, MMBench, MM-Vet, and LLaVA-Bench (in-the-wild)), questions from the annotation files were used as prompts, formatted to fit the chat templates of each respective MLLM. Specifically, we utilized the COCO, A-OKVQA, and GQA datasets for POPE evaluation, and MMBench_DEV_EN for MMBench. In the MM-Vet evaluation, we used an online evaluator powered by OpenAI GPT-4 to assess generated results, while for LLaVA-Bench (in-the-wild) and HallusionBench, we employed OpenAI's model gpt4-1106-preview and GPT-4o-mini respectively via API. For CHAIR, a randomly sampled image set from the COCO Val2014 dataset was used across all three models, with the prompt "Please describe this image in detail." We sampled three different sets of images using different random seeds and evaluated performance by calculating the mean and standard deviation of the results.

All MemVR tests were conducted using a greedy decoding approach, with do_sample=False, temperature=0, threshold=0.75, and beam=1. For VCD tests, we set do_sample=True, temperature=1, noise_step=500, and the plausibility constraint hyperparameter $\lambda$ to 0.1, while $\alpha$, which controls the degree of contrastive emphasis, was set to 1, following the default parameter settings from the original code and literature. OPERA tests were configured with beam=5, sample=True, scale_factor=50, threshold=15, num_attn_candidates=5, and penalty_weights=1. Due to OPERA's reliance on older versions of Torch and Transformers, it was incompatible with Qwen and GLM models, and thus experiments involving these models were not conducted. Additionally, our method introduces two hyperparameters: the informative layer $l$ for activation calculations and the factor $\lambda$ to control the influence of entropy on the next token probability distribution. To map the hidden states from selected layers $l$ to vocabulary tokens, we chose intermediate layers based on the model's depth (e.g., layers 5 to 16 for vicuna-7b, which has 32 layers), and we set $\lambda$ as a fixed value (e.g., 0.75). All parameter settings adhered to the default configurations specified in the respective papers and code repositories.

**Experimental Code**. To promote transparency and ensure the reproducibility of our work, we will release all experimental code, datasets, and detailed tutorials necessary for replicating our experiments. Our goal is to make it straightforward for researchers and practitioners to reproduce our results, regardless of their technical background. Additionally, by providing comprehensive documentation and clear guidelines, we aim to facilitate the extension of our method to other models and architectures, enabling the broader research community to explore its potential applications and improvements. We believe that open and reproducible research is essential for advancing the field and fostering collaboration.

**Computational Resources.** Our experiments were conducted on eight A40 and four A800 GPUs. The computational bottleneck was not the numerical accuracy values but the collection of potential hallucinatory factors for analytical purposes, including logits and attention values for each head and layer.

## B.4 Case Study

This case study aims to evaluate and present various benchmark cases across multiple domains systematically.

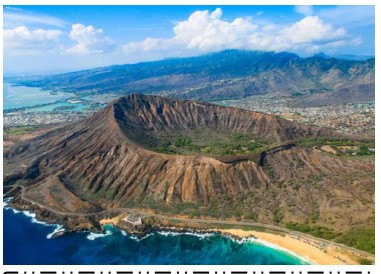

**Ground Truth Caption:** An aerial view of Diamond Head in the Hawaiian Islands.

**Llava-v1.5-7b Default:** The famous sight in the photo is the Na Pali Coast on the island of Kaua'i in Hawaii.

**Llava-v1.5-7b + VCD:** The name of the famous sight in the photo is the Na Pali Coast of Kaua'i Island, which is an island in Hawaii.

**Llava-v1.5-7b Opera:** The name of the famous sight in the photo is the Grand Canyon.

**Llava-v1.5-7b + MemVR (Ours):** The famous sight in the photo is the Diamond Head, which is a volcanic crater located on the island of Oahu in Hawaii.

Question: What is the name of this famous sight in the photo?

Figure 10: A case study comparing the levels of hallucination among various baselines.

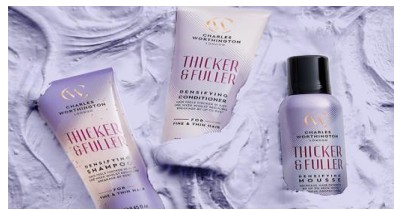

**Ground Truth Caption:** mousse

**Llava-v1.5-7b Default:** The right most bottle is of Charles Worthington's Thicker & Fuller range.

**Llava-v1.5-7b + VCD:** densifying spray

**Llava-v1.5-7b + MemVR (Ours):** The right most bottle is the Thicker & Fuller Volumizing Mousse.

Question: What is the name of this famous sight in the photo?

Figure 11: A case study comparing the levels of hallucination among various baselines.

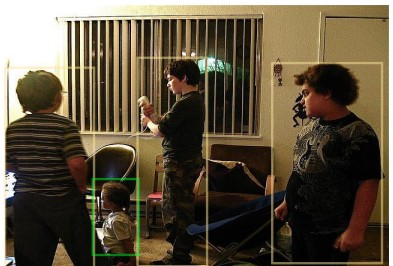

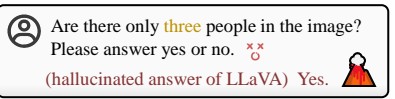

Are there only three people in the image? Please answer yes or no.
(hallucinated answer of LLaVA) Yes.

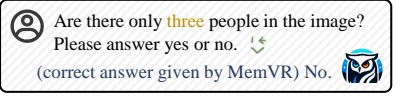

Are there only three people in the image? Please answer yes or no.
(correct answer given by MemVR) No.

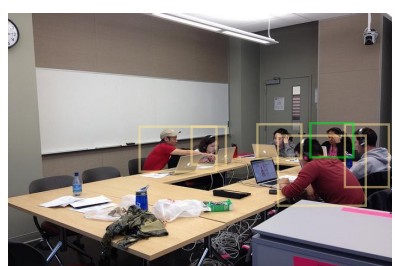

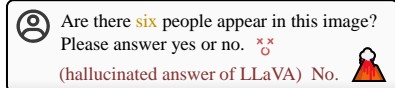

Are there six people appear in this image? Please answer yes or no.
(hallucinated answer of LLaVA) No.

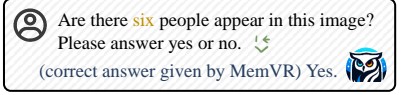

Are there six people appear in this image? Please answer yes or no.
(correct answer given by MemVR) Yes.

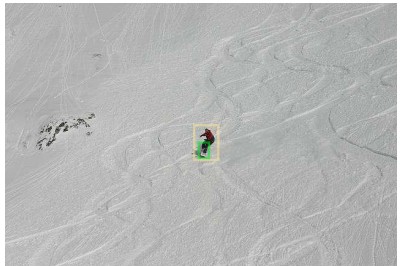

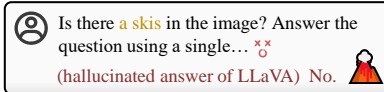

Is there a skis in the image? Answer the question using a single…
(hallucinated answer of LLaVA) No.

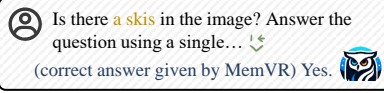

Is there a skis in the image? Answer the question using a single…
(correct answer given by MemVR) Yes.

Figure 12: A case study comparing the levels of hallucination among various baselines.

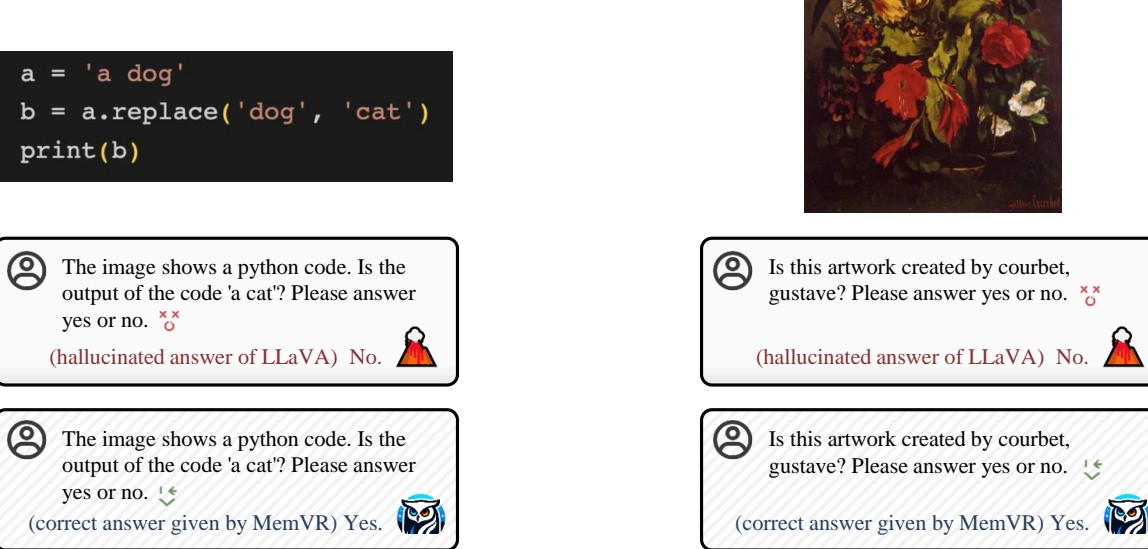

```
a = 'a dog'
b = a.replace('dog', 'cat')
print(b)
```

The image shows a python code. Is the output of the code 'a cat'? Please answer yes or no.

(hallucinated answer of LLaVA)  No.

The image shows a python code. Is the output of the code 'a cat'? Please answer yes or no.

(correct answer given by MemVR) Yes.

Is this artwork created by courbet, gustave? Please answer yes or no.

(hallucinated answer of LLaVA)  No.

Is this artwork created by courbet, gustave? Please answer yes or no.

(correct answer given by MemVR) Yes.

Figure 13: A case study comparing the levels of hallucination among various baselines.

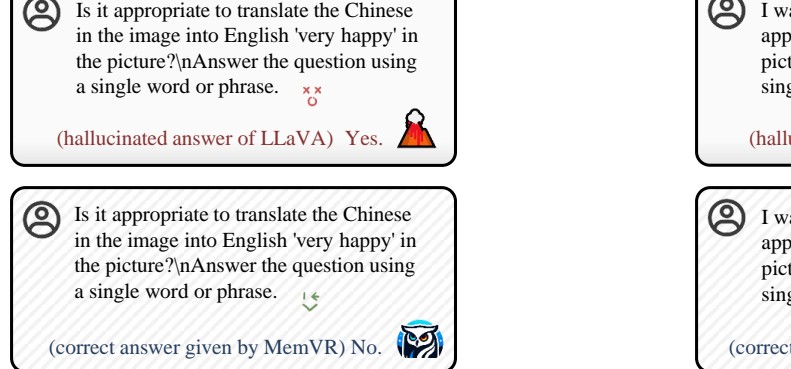

Is it appropriate to translate the Chinese in the image into English 'very happy' in the picture?\nAnswer the question using a single word or phrase.

(hallucinated answer of LLaVA)  Yes.

Is it appropriate to translate the Chinese in the image into English 'very happy' in the picture?\nAnswer the question using a single word or phrase.

(correct answer given by MemVR) No.

I want to supplement protein. Is it appropriate to eat the food in the picture?\nAnswer the question using a single word or phrase.

(hallucinated answer of LLaVA)  No.

I want to supplement protein. Is it appropriate to eat the food in the picture?\nAnswer the question using a single word or phrase.

(correct answer given by MemVR) Yes.

Figure 14: A case study comparing the levels of hallucination among various baselines.

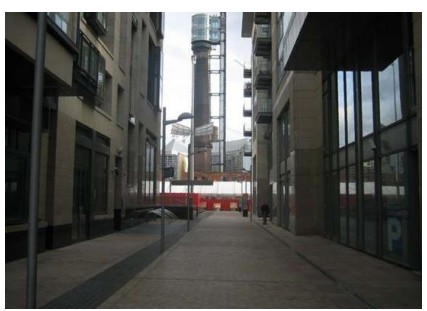

Is this a picture of Smithfield, Dublin?\nAnswer the question using a single word or phrase.

(hallucinated answer of Qwen)  Unknown.

Is this a picture of Smithfield, Dublin?\nAnswer the question using a single word or phrase.

(correct answer given by MemVR) Yes.

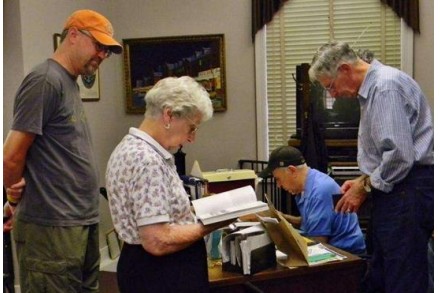

Is this photo taken in a place of nursing home?\nAnswer the question using a single word or phrase.

(hallucinated answer of Qwen)  No.

Is this photo taken in a place of nursing home?\nAnswer the question using a single word or phrase.

(correct answer given by MemVR) Yes.

Figure 15: A case study comparing the levels of hallucination among various baselines.

```
a = 'a dog'
b = a.replace('dog', 'cat')
print(b)
```

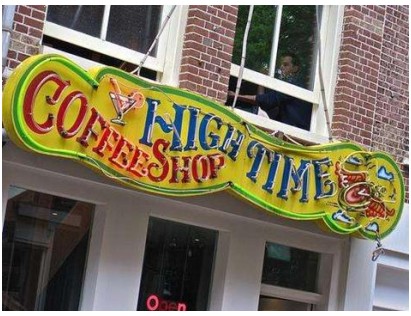

The image shows a python code. Is the output of the code 'a cat'? Please answer yes or no.

(hallucinated answer of Qwen)  No.

The image shows a python code. Is the output of the code 'a cat'? Please answer yes or no.

(correct answer given by MemVR) Yes.

Is the word in the logo \"high tite cofeee shop\"?\nAnswer the question using a single word or phrase.

(hallucinated answer of Qwen)  Yes.

Is the word in the logo \"high tite cofeee shop\"?\nAnswer the question using a single word or phrase.

(correct answer given by MemVR) No.

Figure 16: A case study comparing the levels of hallucination among various baselines.

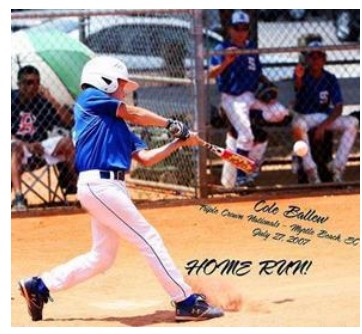

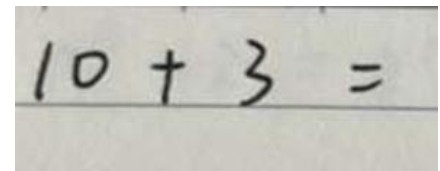

Are there five people in this image? Please answer yes or no.

(hallucinated answer of ChatGLM) Yes.

Is the answer to the arithmetic question in the image 12?\nAnswer the question using a single word or phrase.

(hallucinated answer of ChatGLM) Yes.

Are there five people in this image? Please answer yes or no.

(correct answer given by MemVR) No.

Is the answer to the arithmetic question in the image 12?\nAnswer the question using a single word or phrase.

(correct answer given by MemVR) No.

Figure 17: A case study comparing the levels of hallucination among various baselines.

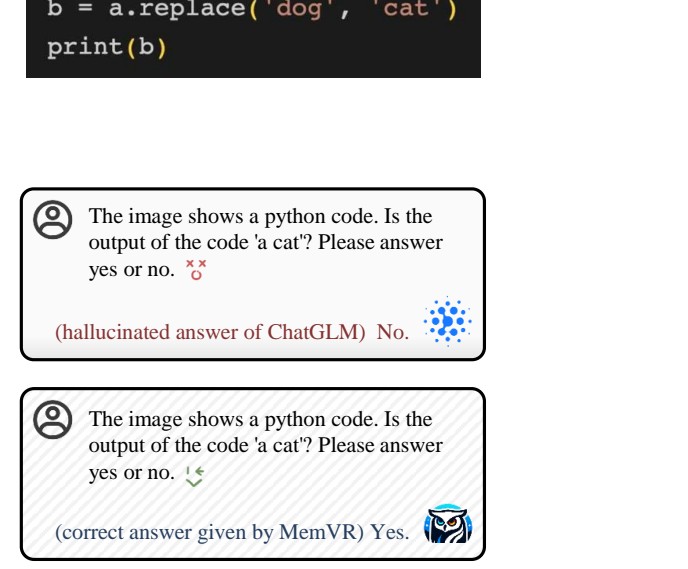

```
a = 'a dog'
b = a.replace('dog', 'cat')
print(b)
```

The image shows a python code. Is the output of the code 'a cat'? Please answer yes or no.

(hallucinated answer of ChatGLM) No.

Is this an image of Sampsonievsky Cathedral?\nAnswer the question using a single word or phrase.

(hallucinated answer of ChatGLM) No.

The image shows a python code. Is the output of the code 'a cat'? Please answer yes or no.

(correct answer given by MemVR) Yes.

Is this an image of Sampsonievsky Cathedral?\nAnswer the question using a single word or phrase.

(correct answer given by MemVR) Yes.

Figure 18: A case study comparing the levels of hallucination among various baselines.

## B.5 Additional Experiments and Results

| Method | MME | | | | | |
| --- | --- | --- | --- | --- | --- | --- |
| | Overall | | Perception | | Cognition | |
| LLaVA1.5-7B | 1864.68 | ↑0.0 | 1508.97 | ↑0.0 | 355.71 | ↑0.0 |
| + VCD (Leng et al., 2024) | 1872.87 | ↑8.2 | 1515.01 | ↑6.0 | 357.86 | ↑2.2 |
| + OPERA (Huang et al., 2024a) | 1784.34 | ↓80.3 | 1473.62 | ↓35.3 | 310.71 | ↓45.0 |
| + ICD (Wang et al., 2024) | 1594.77 | ↓269.9 | 1306.91 | ↓202.1 | 287.86 | ↓67.9 |
| + MemVR (Ours) | 1896.72 | ↑32.0 | 1512.80 | ↑3.8 | 383.92 | ↑28.2 |
| Qwen-VL-Chat | 1784.93 | ↑0.0 | 1442.79 | ↑0.0 | 342.14 | ↑0.0 |
| + VCD (Leng et al., 2024) | 1721.06 | ↓63.9 | 1403.17 | ↓39.6 | 317.89 | ↓24.3 |
| + OPERA (Huang et al., 2024a) | - | | - | | - | |
| + ICD (Wang et al., 2024) | 1833.75 | ↑48.8 | 1472.06 | ↑29.3 | 361.69 | ↑19.6 |
| + MemVR (Ours) | 1820.95 | ↑36.0 | 1473.45 | ↑30.7 | 347.50 | ↑5.4 |
| GLM-4V-9B | 2160.54 | ↑0.0 | 1680.89 | ↑0.0 | 479.64 | ↑0.0 |
| + VCD (Leng et al., 2024) | 2105.53 | ↓55.0 | 1624.10 | ↓56.8 | 481.43 | ↑1.8 |
| + OPERA (Huang et al., 2024a) | - | | - | | - | |
| + ICD (Wang et al., 2024) | 2074.30 | ↓86.2 | 1566.09 | ↓114.8 | 508.21 | ↑28.6 |
| + MemVR (Ours) | 2170.17 | ↑9.6 | 1683.03 | ↑2.1 | 487.14 | ↑7.5 |

Table A1: Results on the MME dataset. Best-performing method per model size is in bold. Arrows indicate improvement ( ↑) or degradation ( ↓) vs. the baseline model.

| Method | LLaVABench (in-the-wild) | | | | | | | |
| --- | --- | --- | --- | --- | --- | --- | --- | --- |
| | Average | | All_1 | | All_2 | | All_3 | |
| LLaVA1.5-7B | 64.80 | ↑0.0 | 63.40 | ↑0.0 | 80.20 | ↑0.0 | 50.80 | ↑0.0 |
| + VCD (Leng et al., 2024) | 63.20 | ↓1.6 | 59.10 | ↓4.3 | 82.00 | ↑1.8 | 48.50 | ↓2.3 |
| + OPERA (Huang et al., 2024a) | 64.30 | ↓0.5 | 59.80 | ↓3.6 | 83.30 | ↑3.1 | 49.80 | ↓1.0 |
| + ICD (Wang et al., 2024) | 56.90 | ↓7.9 | 49.80 | ↓13.6 | 80.70 | ↑0.5 | 40.20 | ↓10.6 |
| + MemVR (Ours) | 65.17 | ↑0.4 | 64.00 | ↑0.6 | 80.20 | ↑0.0 | 51.30 | ↑0.5 |
| Qwen-VL-Chat | 68.50 | ↑0.0 | 70.40 | ↑0.0 | 79.30 | ↑0.0 | 55.80 | ↑0.0 |
| + VCD (Leng et al., 2024) | 53.77 | ↓14.7 | 41.00 | ↓29.4 | 85.30 | ↑6.0 | 35.00 | ↓20.8 |
| + OPERA (Huang et al., 2024a) | - | | - | | - | | - | |
| + ICD (Wang et al., 2024) | 56.60 | ↓11.9 | 45.30 | ↓25.1 | 85.70 | ↑6.4 | 38.80 | ↓17.0 |
| + MemVR (Ours) | 69.50 | ↑1.0 | 69.50 | ↓0.9 | 82.00 | ↑2.7 | 57.00 | ↑1.2 |
| GLM-4V-9B | 75.30 | ↑0.0 | 88.40 | ↑0.0 | 73.00 | ↑0.0 | 64.50 | ↑0.0 |
| + VCD (Leng et al., 2024) | 74.23 | ↓1.1 | 86.70 | ↓1.7 | 72.80 | ↓0.2 | 63.20 | ↓1.3 |
| + OPERA (Huang et al., 2024a) | - | | - | | - | | - | |
| + ICD (Wang et al., 2024) | 70.97 | ↓4.3 | 82.10 | ↓6.3 | 71.80 | ↓1.2 | 59.00 | ↓5.5 |
| + MemVR (Ours) | 76.73 | ↑1.4 | 88.90 | ↑0.5 | 74.80 | ↑1.8 | 66.50 | ↑2.0 |

Table A2: Results on LLaVABench (in-the-wild) dataset. Best-performing method per model size and dataset is highlighted in bold; arrows indicate improvement or degradation over the baseline, where higher values indicate better performance.

| Method | Total |
|---|---|
| LLaVA1.5-7B | 31.1 ↑0.0 |
| + VCD (Leng et al., 2024) | 30.2 ↓0.9 |
| + OPERA (Huang et al., 2024a) | 32.0 ↑0.9 |
| + ICD (Wang et al., 2024) | 25.9 ↓5.2 |
| + MemVR (Ours) | 32.4 ↑1.3 |
| Qwen-VL-Chat | 49.0 ↑0.0 |
| + VCD (Leng et al., 2024) | 34.6 ↓14.4 |
| + OPERA (Huang et al., 2024a) | - |
| + ICD (Wang et al., 2024) | 31.7 ↓17.3 |
| + MemVR (Ours) | 49.6 ↑0.6 |
| GLM-4V-9B | 63.4 ↑0.0 |
| + VCD (Leng et al., 2024) | 59.4 ↓4.0 |
| + OPERA (Huang et al., 2024a) | - |
| + ICD (Wang et al., 2024) | 57.7 ↓5.7 |
| + MemVR (Ours) | 65.0 ↑1.6 |

Table A3: Results on MM-Vet dataset. Best-performing method per model size is highlighted in bold. Arrows indicate improvement ( ↑) or degradation ( ↓) over the baseline.

| Method | Accuracy |
|---|---|
| LLaVA1.5-7B | 50.00 ↑0.0 |
| + VCD (Leng et al., 2024) | 44.90 ↓5.1 |
| + OPERA (Huang et al., 2024a) | 50.76 ↑0.8 |
| + ICD (Wang et al., 2024) | 37.62 ↓12.4 |
| + MemVR (Ours) | 51.50 ↑1.5 |
| Qwen-VL-Chat | 66.05 ↑0.0 |
| + VCD (Leng et al., 2024) | 34.54 ↓31.5 |
| + OPERA (Huang et al., 2024a) | - |
| + ICD (Wang et al., 2024) | 29.37 ↓36.7 |
| + MemVR (Ours) | 66.36 ↑0.3 |
| GLM-4V-9B | 57.39 ↑0.0 |
| + VCD (Leng et al., 2024) | 48.04 ↓9.4 |
| + OPERA (Huang et al., 2024a) | - |
| + ICD (Wang et al., 2024) | 50.01 ↓7.4 |
| + MemVR (Ours) | 58.00 ↑0.6 |

Table A4: Results on the Vizwiz dataset. Best-performing method per model size is highlighted in bold; arrows indicate improvement ( ↑) or degradation ( ↓) relative to the baseline.

| Method | CHAIRS | | | |
|---|---|---|---|---|
| | Cs | Ci | Recall | Len |
| LLaVA1.5-7B | 47.60 ↑0.0 | 13.30 ↑0.0 | 80.60 ↑0.0 | 99.70 ↑0.0 |
| + VCD (Leng et al., 2024) | 55.00 ↑7.4 | 15.80 ↑2.5 | 77.40 ↓3.2 | 101.20 ↑1.5 |
| + OPERA (Huang et al., 2024a) | 47.60 ↑0.0 | 13.50 ↑0.2 | 79.00 ↓1.6 | 93.20 ↓6.5 |
| + ICD (Wang et al., 2024) | 56.20 ↑8.6 | 16.30 ↑3.0 | 16.30 ↓64.3 | 103.40 ↑3.7 |
| + MemVR (Ours) | 46.60 ↓1.0 | 13.00 ↓0.3 | 80.80 ↑0.2 | 99.60 ↓0.1 |
| Qwen-VL-10B | 6.80 ↑0.0 | 5.30 ↑0.0 | 53.40 ↑0.0 | 17.60 ↑0.0 |
| + VCD (Leng et al., 2024) | 13.00 ↑6.2 | 12.30 ↑7.0 | 47.90 ↓5.5 | 115.70 ↑98.1 |
| + OPERA (Huang et al., 2024a) | - | - | - | - |
| + ICD (Wang et al., 2024) | 18.40 ↑11.6 | 14.30 ↑9.0 | 37.60 ↓15.8 | 48.10 ↑30.5 |
| + MemVR (Ours) | 4.80 ↓2.0 | 3.30 ↓2.0 | 52.30 ↓1.1 | 15.00 ↓2.6 |
| GLM-4V-9B | 40.40 ↑0.0 | 9.00 ↑0.0 | 72.70 ↑0.0 | 218.20 ↑0.0 |
| + VCD (Leng et al., 2024) | 42.20 ↑1.8 | 9.60 ↑0.6 | 72.80 ↑0.1 | 239.80 ↑21.6 |
| + OPERA (Huang et al., 2024a) | - | - | - | - |
| + ICD (Wang et al., 2024) | 43.40 ↑3.0 | 9.10 ↑0.1 | 73.60 ↑1.9 | 239.80 ↑21.6 |
| + MemVR (Ours) | 39.40 ↓1.0 | 9.00 ↑0.0 | 70.70 ↓2.0 | 214.00 ↓4.2 |

Table A5: Results on CHAIRS dataset. Best-performing method per model size and dataset is highlighted in bold; arrows indicate improvement or degradation over the baseline, where lower values indicate better performance.

| Method | Existence | Count | Position | Color | Scene | Artwork | OCR | Numerical | Text_trans | Code_reason |
|---|---|---|---|---|---|---|---|---|---|---|
| LLaVA-Next (Llama3-8B) | 195.0 | 165.0 | 143.3 | 185.0 | 161.6 | 159.2 | 118.0 | 125.0 | 50.0 | 77.5 |
| +MemVR | 195.0 | 170.0 | 143.3 | 185.0 | 163.6 | 161.0 | 124.0 | 125.0 | 52.5 | 77.5 |
| LLaVA-Next (Mistral-7B) | 190.0 | 150.0 | 133.3 | 190.0 | 144.2 | 163.5 | 113.0 | 122.5 | 60.0 | 67.5 |
| +MemVR | 195.0 | 155.0 | 133.3 | 190.0 | 145.2 | 165.0 | 113.8 | 122.5 | 60.0 | 67.5 |
| LLaVA-Next (Vicuna-1.6-7B) | 195.0 | 135.0 | 143.3 | 165.0 | 162.2 | 123.2 | 132.5 | 42.5 | 107.5 | 55.0 |
| +MemVR | 195.0 | 135.0 | 135.0 | 170.0 | 163.0 | 123.5 | 140.0 | 42.5 | 115.0 | 57.5 |

Table A6: Performance comparison across different LLaVA-Next models with and without MemVR.

| Method | MMBench | | | | | | |
|---|---|---|---|---|---|---|---|
| | AR | CP | FP-C | FP-S | LR | RR | Overall |
| LLaVA1.5-7B | 72.86 ↑0.0 | 75.68 ↑0.0 | 58.04 ↑0.0 | 63.48 ↑0.0 | 28.81 ↑0.0 | 51.30 ↑0.0 | 62.80 ↑0.0 |
| + VCD (Leng et al., 2024) | 60.30 | 68.58 | 51.75 | 53.24 | 18.64 | 48.70 | 54.21 |
| + OPERA (Huang et al., 2024a) | 69.85 | 75.00 | 56.64 | 66.21 | 28.81 | 53.04 | 62.80 |
| + ICD (Wang et al., 2024) | 42.21 ↓30.7 | 52.36 ↓23.3 | 52.36 ↓5.7 | 39.59 ↓23.9 | 16.10 ↓12.7 | 32.17 ↓19.1 | 39.78 ↓23.0 |
| + MemVR (Ours) | 71.86 ↑1.2 | 76.69 ↑1.0 | 57.34 ↓0.7 | 64.16 ↑0.9 | 31.36 ↑2.5 | 56.52 ↑5.2 | 63.75 ↑0.9 |
| Qwen-VL-10B | 60.30 ↑0.0 | 71.28 ↑0.0 | 45.45 ↑0.0 | 62.80 ↑0.0 | 28.81 ↑0.0 | 38.26 ↑0.0 | 56.53 ↑0.0 |
| + VCD (Leng et al., 2024) | 34.67 | 52.36 | 20.28 | 55.63 | 11.86 | 22.61 | 39.18 |
| + OPERA (Huang et al., 2024a) | - | - | - | - | - | - | - |
| + ICD (Wang et al., 2024) | 12.56 ↓47.7 | 17.57 ↓53.7 | 2.10 ↓43.4 | 22.53 ↓40.3 | 2.54 ↓26.3 | 5.22 ↓33.0 | 13.32 ↓43.2 |
| + MemVR (Ours) | 61.31 ↑1.0 | 71.28 ↑0.0 | 44.06 ↓1.4 | 62.80 ↑0.0 | 27.97 ↓0.8 | 38.26 ↑0.0 | 56.44 ↓0.1 |
| GLM-4V-9B | 88.44 ↑0.0 | 86.49 ↑0.0 | 69.93 ↑0.0 | 85.67 ↑0.0 | 66.10 ↑0.0 | 85.22 ↑0.0 | 82.39 ↑0.0 |
| + VCD (Leng et al., 2024) | 86.43 | 85.47 | 68.53 | 84.64 | 61.86 | 81.74 | 80.58 |
| + OPERA (Huang et al., 2024a) | - | - | - | - | - | - | - |
| + ICD (Wang et al., 2024) | 83.92 ↓4.5 | 84.46 ↓2.0 | 62.94 ↓7.0 | 81.23 ↓4.4 | 60.17 ↓5.9 | 80.00 ↓5.2 | 78.01 ↓4.4 |
| + MemVR (Ours) | 88.94 ↑0.5 | 86.49 ↑0.0 | 70.63 ↑0.7 | 86.01 ↑0.4 | 66.10 ↑0.0 | 85.22 ↑0.0 | 82.65 ↑0.3 |

Table A7: Results on the MMBench dataset with newly added ICD rows. Best-performing method per model size and dataset is highlighted in bold; arrows indicate improvement ( ↑) or degradation ( ↓) over the baseline.

| Strategy | 1-Token Len | 5-Token Len | 10-Token Len | 20-Token Len | 30-Token Len | 50-Token Len | 80-Token Len |
|---|---|---|---|---|---|---|---|
| Greedy | 661.7 | 897.9 | 1273.1 | 1880.3 | 2501.8 | 3617.6 | 5256.6 |
| Sample | 786.8 | 1056.2 | 1314.9 | 1998.5 | 2568.5 | 3593.0 | 5587.0 |
| VCD Sample | 1747.74 | 2767.52 | 4027.07 | 4537.42 | 5031.39 | 7690.77 | 11569.3 |
| Opera Beam | 1566.1 | 3094.9 | 4166.4 | 6242.7 | 8436.9 | 12672.3 | 19247.2 |
| MemVR Sample | 750.8 | 1197.6 | 1780.5 | 2339.2 | 2631.7 | 3718.0 | 6011.0 |
| MemVR Greedy | 775.1 | 974.2 | 1337.5 | 1861.7 | 2742.8 | 4000.9 | 5545.5 |

Table A8: Time cost for generating tokens. All based on LLaVA1.5-7B

## B.6 VCD Implement Details

The code of VCD (Leng et al., 2024) is also released. However, the result of VCD evaluated in our experiments on POPE is lower than the original paper. Therefore, we report the results in the original paper.

# C  Examples of capability integrations

## C.1  Others

*Q: How can the model understand information directly from the vision encoder, especially if it has a different vision system?*
To ensure that MemVR is adaptable across diverse vision systems, we conducted experiments on multiple VLM architectures, including LLaVA, which utilizes a Visual-Instructional-Tuning framework with different sizes of ViT-based CLIP models, Qwen-VL-Chat, which employs a Q-Former-like architecture for visual processing, and ChatGLM-4v-9B, which integrates a large pre-trained visual encoder. These architectures encompass a broad range of vision models, providing confidence that MemVR is applicable to most MLLMs in use today.

**Artifacts and licenses** We report a list of licenses for all datasets and models used in our experiment in Table A10. We strictly follow all the model licenses and limit the scope of these models to academic research only.

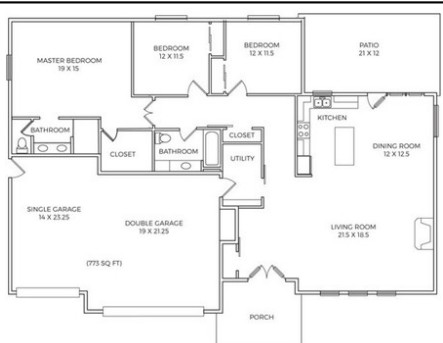

**Question:** Which room is bigger, the double garage or the living room?
**Ground Truth:** Double garage
**Required Capabilities:** OCR, Spatial Awareness, Math

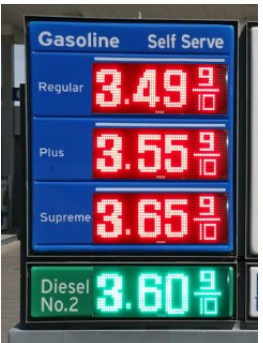

**Question:** How many gallons of supreme gasoline can I get with $50?
**Ground Truth:** 13.6 | 13.7
**Required capabilities:** OCR, Math

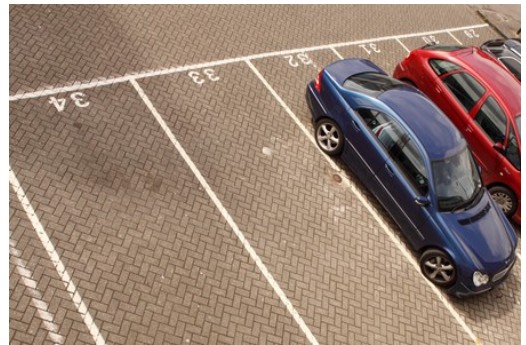

**Question:** Which car is on the parking spot 33?
**Ground Truth:** No | Empty
**Required Capabilities:** Recognition, OCR, Spatial Awareness

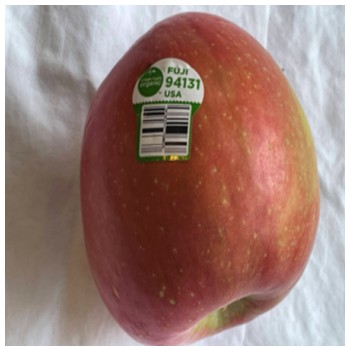

**Question:** Is this apple organic?
**Ground Truth:** Yes
**Required capabilities:** Recognition, OCR

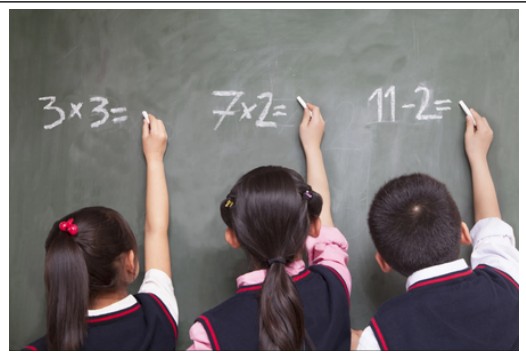

**Question:** What will the girl on the right write on the board?
**Ground Truth:** 14
**Required capabilities:** Recognition, OCR, Spatial Awareness, Math

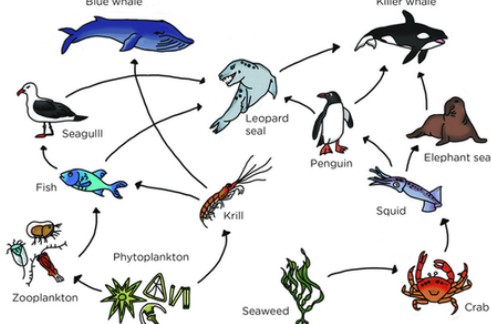

**Question:** Which are producers in this food web?
**Ground Truth:** Phytoplankton & Seaweed
**Required Capabilities:** OCR, Knowledge, Spatial Awareness

Table A9: Six samples on MM-Vet benchmark requiring different capability integrations.

| Data Sources | URL | License |
|---|---|---|
| MSCOCO 2017 | Link | CC BY 4.0 |
| ADE20K | Link | BSD-3-Clause |
| VQA Val | Link | CC BY 4.0 |
| LLaVA-bench-in-the-wild | Link | Apache-2.0 |
| ImageNet | Link | Custom License |
| MMBench | Link | Apache-2.0 |
| **Software Code** | **URL** | **License** |
| LLaVA | Link | Llama Community Licence |
| Qwen-VL | Link | Tongyi Qianwen Licence |
| GLM-4V | Link | THUDM GLM-4 Licence |
| GPT-4V/4O | Link | OpenAI Term of Use |

Table A10: License information for the scientific artifacts.

