# OpenReview forum: "Look Twice Before You Answer: Memory-Space Visual Retracing for Hallucination Mitigation in Multimodal Large Language Models"
_ICML.cc/2025/Conference — ICML 2025 poster_

### Official Review · Reviewer_8UAJ · 2025-03-10

**Overall Recommendation:** 3

**Summary:**

The paper addresses object hallucination, a well-known issue for existing multimodal large language models (MLLMs), where they prone to generate plausible yet incorrect responses that are not aligned with given images. Authors attribute this to weak robustness and high uncertainty of LLM representations (section 3.2), which they refer to as “amnesia” of MLLMs like LLaVA. From this perspective, authors propose to enhance representations by retracing.

**Claims And Evidence:**

Some claims made in this submission are not supported well. A few things should be taken care of more meticulously. To reveal amnesia phenomenon of existing MLLMs, authors pass representations from all layers in LLaMA to a vocabulary head, which is aligned with the last layer solely during LLaMA pre-training. Given that most layers are not aligned with the vocabulary head, what such experiments indicate is doubtful and the conclusions do not make sense. This is problematic and misleading analysis. Another thing is, authors name this to amnesia and memory, however, I do not see memory-involved design in their approach. Meanwhile, authors seem confuse the difference between hallucination and error. Specifically, the Figure 5 case is not a clear hallucination problem, which is not a language prior or statistical bias.

**Essential References Not Discussed:**

The visual retracing appears to be similar to V*, where authors should discuss common ideas and differences more clearly, along with its follow-up studies.

[a] Guided Visual Search as a Core Mechanism in Multimodal LLMs.

**Experimental Designs Or Analyses:**

The experimental designs or analyses has weak relation to experimental designs.

**Methods And Evaluation Criteria:**

The proposed method is not sensical from technical perspective, as most layers are not aligned with the vocabulary head during pre-training, the analysis in section 3 is problematic.

**Other Comments Or Suggestions:**

No other comments so far.

**Other Strengths And Weaknesses:**

No other strengths and weaknesses so far.

**Questions For Authors:**

No questions for now.

**Relation To Broader Scientific Literature:**

The proposed method is technically viable to be applied at a general task than object hallucination, such as MMBench and MMStar.

**Theoretical Claims:**

No theoretical claims are made in main paper.

---

> ### Author Rebuttal · Authors · 2025-03-30
>
> **We sincerely thank Reviewer 8UAJ for the insightful comments.**
>
> > Q1: Given that most layers are not aligned with the vocabulary head, what such experiments indicate is doubtful and the conclusions do not make sense...
>
> The idea of applying language heads directly to the hidden states of the middle layers, known as **early exit** [1][2][3], has proven to be effective even without the special training process [4][5] for alignment to the vocabulary head.
>
> We also presented it in the paper (*Line 263, right*).
>
> [1] Teerapittayanon et al. Branchynet: Fast inference via early exiting from deep neural networks. ICPR, 2016.
>
> [2] Maha Elbayad et al. Depth-adaptive transformer. ICLR, 2020.
>
> [3] Tal Schuster et al. Confident adaptive language modeling. NeurIPS, 2022.
>
> [4] Wei-Tsung Kao et al. Bert's output layer recognizes all hidden layers? some intriguing phenomena and a simple way to boost bert. arXiv, 2020.
>
> [5] Chuang Y S et al. DoLa: Decoding by Contrasting Layers Improves Factuality in Large Language Models. ICLR 2024.
>
> > Q2: About 'amnesia' and not seeing memory-involved design
>
> "Amnesia" is similar to the concept of the information flow of image tokens converges at shallow layers and diverges at deeper layers in [6], where truncating image tokens no longer affects the answers in the middle or deep layers.
>
> VR is implemented in FFN, and FFN stores knowledge from training data in the form of KV memories [7], thus we call VR in memory space.
>
> [6] From Redundancy to Relevance: Enhancing Explainability in Multimodal Large Language Models, 2024.
>
> [7] Geva M, et al. Transformer Feed-Forward Layers Are Key-Value Memories. EMNLP, 2021.
>
> > Q3: Figure 5 case is not a clear hallucination problem, which is not a language prior or statistical bias
>
> The ground-truth of Figure 5 case is "*four* **mangosteen**", but LLaVA outputs "*a* **pomegranate**", the count (four->one) and object (mangosteen->pomegranate) are wrong. We don't know why it is not a clear hallucination case.
>
> 1) About this case, Reviewer 8UAJ thinks it is not a language prior or statistical bias. We get the reviewer's view that only examples with a language prior or statistical bias, like the misidentification of a black banana as a yellow banana, can be called hallucinations. **However, the language prior or statistical bias is merely a possible source of the hallucinations** proposed in the Contrast Decoding series of papers, e.g., VCD, but not the only source.
>
> 2) We're not imitating VCD or other papers to present a language prior or statistical bias as the cause of hallucination, so we did not purposely choose like “black bananas“ with a strong language prior for this example.
>
> 3) Further, **the other hallucination cases with a language prior or statistical bias in our experiments also show the same regularity, i.e., hallucination tokens exhibit higher entropy than correct ones.**
>
> > Q4: No theoretical claims
>
> We have provided a theoretical framework based on information theory in the paper, **Reviewers 2zfj,bEEH,3FtN mentioned**.
>
> > Q5: The experimental designs or analyses has weak relation to experimental designs
>
> Firstly, we actually don't know much about what you mean by weak ties.
>
> We conducted comprehensive experiments and analysis on 8 benchmarks and ablation studies examining the impact of different injection ratios, uncertainty thresholds, and static vs. dynamic triggering strategies.
>
> We've already provided 5 MLLMs with 8 kinds of benchmark evaluations. These MLLMs used different Language models, text-image aligning methods, and training strategies to ensure effectiveness and generalization across the various testing environments. And, we have demonstrated that MemVR can be easily applied to almost all open-source MLLMs. Compared with SOTA methods, we have done the most complete experiments, as follows, ﻿ ﻿
>
> |Method|Evaluation benchmarks|
> |--------|:--------|
> |OPERA|POPE, MME, CHAIR, MMBench (4)|
> |HALC|POPE, MME, LLaVA-Bench (3)|
> |VCD|POPE, MME, LLaVA-Bench (3)|
> |ICD|POPE, MME, LLaVA-Bench (3)|
> |MemVR (ours) |POPE, MME, LLaVA-Bench, CHAIR, HallusionBench, MMBench, MM-Vet,  VizWiz-VQA **(8)**|
>
> **If you still think our experimental design has problems, could you be more specific? We are willing to discuss with you.**
>
> > Q6: VR appears to be similar to V*, about the differences
>
> We actually don't understand why Reviewer 8UAJ suddenly gave us a paper that is irrelevant to hallucinations, nor in a similar field. Nonetheless, **we follow your comment for comparison.**
>
> **In short, the difference between V\* and ours is as follows,**
>
> 1) The goal is different. V* is an LLM-guided visual search algorithm, not a method to mitigate hallucination.
>
> 2)  The framework of V* is the normal structure of MLLMs, which is completely different from our "look-twice" where an extra bypass is created.
>
> 3) V* needs training by LoRA, but MemVR is a plug-and-play method.
>
> [1] V*: Guided Visual Search as a Core Mechanism in Multimodal LLMs.

---

> > ### Comment · Reviewer_8UAJ · 2025-04-02
> >
> > I thank authors for such detailed responses in their rebuttals.
> >
> > The claims made by authors have addressed most of my concerns, including Q1 where related works are clearly mentioned. The overview framework has now become clearer to me. I am raising my score to weak accept.

---

### Official Review · Reviewer_3FtN · 2025-03-11

**Overall Recommendation:** 3

**Summary:**

This paper presents Memory-space Visual Retracing (MemVR), a decoding strategy aimed at mitigating hallucinations in Multimodal Large Language Models (MLLMs). The primary insight is that MLLMs tend to lose visual information during the decoding process, leading to hallucinations due to an over-reliance on textual context. Inspired by human cognition, MemVR introduces a “look-twice” mechanism, where visual tokens are re-injected into intermediate transformer layers based on model uncertainty. This approach leverages the Feed Forward Network (FFN) as a key-value memory module, supplementing missing visual information dynamically.MemVR mitigates hallucinations by balancing modality contributions.

**Claims And Evidence:**

Claims:

MemVR mitigates hallucinations by balancing modality contributions.
MemVR is computationally efficient, introducing minimal latency.
These claims are supported from the experiments results.

However, uncertainty-based heuristics may be brittle, as textual uncertainty higher than a fixed threshold does not always indicate a hallucination problem. The ablation in fig.7 (left) doesn't show the usefulness of the threshold.

**Essential References Not Discussed:**

No

**Experimental Designs Or Analyses:**

Yes.

Comparisons across multiple benchmarks.

Different uncertainty thresholds (γ) and their impact.

Analysis of hallucination uncertainty per layer.

**Methods And Evaluation Criteria:**

Yes

**Other Comments Or Suggestions:**

See weakness

**Other Strengths And Weaknesses:**

#### **Strengths**
- The paper is well-organized and easy to follow, with a logical flow of ideas.
- The study provides a structured analysis of why VLMs hallucinate, supported by attention visualizations and informative figures.
- MemVR achieves good performance on the benchmarks tested, outperforming prior hallucination mitigation methods.

---

#### **Weaknesses**
- **Limited novelty in the analysis of hallucinations**
  - The discussion in Section 3.2 about VLMs failing to correctly process images is already a well-established issue in the field. The analysis does not provide particularly novel insights beyond existing consensus.
  - Figure 4 lacks clarity regarding "text feature"—does this refer to the text prompt’s feature representation?
  - Since VLMs are inherently trained to answer text-based questions, they naturally rely on textual features. A decrease in text feature importance may lead to text with issues such as incoherence, redundancy.

- **Uncertain about the effectiveness of the method and potential bias amplification**
  - Equation (6) injects additional image information, but does this also introduce image-related bias?
  - Prior works, such as [1], highlight how image-biased hallucination can degrade response quality.
  - [2] discusses how image redundancy can lead to errors. Does MemVR amplify such errors?
  - Why not using the uncertainty score as $\alpha$?
  - A more thorough analysis or potential solution for these problems would strengthen the paper.

- **tested benchmarks**
  - While the proposed method appears simple, its contribution lacks significant novelty in the VLH domain.
  - Given the concerns outlined above, is MemVR overfitting to specific benchmarks rather than providing a robust, generalizable solution? The evaluation could be improved by testing MemVR on more recent and challenging hallucination benchmarks, such as HallusionBench [3], or other reasonable ones.

- **Minor issues and typos**
  - Figure 4 and some others contain a typo in the word "perception".

---

#### **References**
**[1]** *IBD: Alleviating Hallucinations in Large Vision-Language Models via Image-Biased Decoding*
**[2]** *From Redundancy to Relevance: Enhancing Explainability in Multimodal Large Language Models*
**[3]** *HallusionBench: An Advanced Diagnostic Suite for Entangled Language Hallucination and Visual Illusion in Large Vision-Language Models*

**Questions For Authors:**

See weakness

**Relation To Broader Scientific Literature:**

The paper situates MemVR well within multimodal hallucination mitigation literature, contrasting:

Retrieval-Augmented Generation

Contrastive Decoding

Attention-based hallucination correction

**Theoretical Claims:**

Yes. The authors provide an information-theoretic justification, showing that MemVR increases mutual information between hidden states and visual tokens, leading to a decrease in hallucinations.

---

> ### Author Rebuttal · Authors · 2025-03-30
>
> **We sincerely thank Reviewer 3FtN for the constructive comments on our work. We promise to revise the paper.**
>
> > Q1: About textual uncertainty
>
> Yes, textual uncertainty cannot be a sufficiently necessary condition for the existence of hallucinations, but when hallucinations are occurring, textual uncertainty tends to be above the threshold, and MemVR triggers override the cases in which hallucinations occur.
>
> Besides, regarding the ablation in Fig. 7 (left), the curve may not be obvious due to the value being over 1800 and the change being within 100. Specificaly, we set $\gamma$ from 0.5 to 1.0, when $\gamma$ is set to less 0.6 MemVR is triggered early without performance improvement, while **between 0.6 and 0.95 can improve performance, with the optimal threshold around 0.75 (can increase 32 scores)**, which means MemVR works well under higher uncertainty.
>
> > Q2: The analysis in Section 3.2 not beyond
>
> There are many works that have discussed why MLLM generates hallucinations. Thus, our discussion in Section 3.2 firstly introduces previous insights, including inherent biases in the training data, visual uncertainty resulting from the model’s statistical bias and priors, and the limitations of current models in accurately discerning context and fact throughout the output generation process.
>
> Then, we presented our argument that the imbalance of modalities leads to a substantial deviation from the accurate representation of visual input, eventually giving rise to hallucinations, which prior work did not raise. We will continue our efforts to delve into the causes of hallucinations.
>
> > Q3: Regarding "text feature"
>
> The 'text feature' here refers to the embedded text tokens, which are concatenated with visual tokens to form the input of LLMs.
>
> > Q4: Does VR introduce image-related bias?
>
> [1] constructs the image-biased model by adjusting the attention weight matrices (i.e., an amplification to the image tokens) within the vanilla model.
>
> In Equation (6), **VR only modifies the hidden state in the FFN layer (i.e., memory about visual information) that uses the original hidden state as query to search supplemental visual feature**, rather than a direct amplification to the image tokens in attention weight matrices, thus VR does not introduce image-related bias as [1].
>
> [1] IBD: Alleviating Hallucinations in Large Vision-Language Models via Image-Biased Decoding
>
> > Q5: Does MemVR amplify image redundancy errors?
>
> In conclusion of [2], image tokens are highly redundant after the cliff layer, where truncating image tokens no longer affects the answers in the middle or deep layers. **This is because LLMs are more text-informed with "attention sinks", and token embedding of image tokens may not align well with the model’s text-based training.** This does not mean that image redundancy leads to errors.
>
> In MemVR, VR supplements visual information to the middle or deep layers of LLMs by taking the original hidden state as queries to search for visual features that need to be supplemented, without introducing image-related bias, nor amplifying image redundancy errors. Experimental results on 8 benchmarks show that VR effectively improves performance.
>
> [2] From Redundancy to Relevance: Enhancing Explainability in Multimodal Large Language Models
>
> > Q6: Why not use the uncertain score as $\alpha$?
>
> Thank your nice suggestion. The uncertain score is usually between 0.5 and 0.99, which is large, and the original knowledge would be confused if we use the uncertain score as $\alpha$. To achieve a dynamic injecting ratio, we calculate $\alpha$ by 2*(uncertain_score – threshold), which this named MemVR++, and the results are as follows,
>
> LLaVA-Bench:
>
> |Method|average|all|complex|conv|detail|
> |--|:--:|:--:|:--:|:--:|:---:|
> |LLaVA1.5|64.80|50.80|74.60|52.90|52.10|
> |VCD|63.20|48.50|77.90|52.40|50.80|
> |ICD|56.90|40.20|78.20|35.30|42.20|
> |MemVR|65.17|51.30|77.90|55.90|52.60|
> |MemVR++|65.87|51.70|81.80|51.20|49.60|
>
> POPE (MSCOCO):
>
> |Method|Random|Popular|Adversarial|Average|
> |--|:--:|:--:|:--:|:--:|
> |LLaVA1.5|83.49|79.98|76.03|79.83|
> |MemVR|88.50|87.10|85.20|86.93|
> |MemVR++|88.40|87.17|85.17|86.92|
>
> The results show that, compared with MemVR, MemVR++ can also achieve the same improvement, and even better.
>
> > Q7: Tested benchmarks and contribution
>
> Follow your suggestion, we test MemVR on the challenging **HallusionBench**, the results are as follows,
>
> |Method|$fAcc$|$easy aAcc$|$hard aAcc$|$aAcc$|
> |--|:--:|:--:|:--:|:--:|
> |LLaVA1.5|17.92|36.04|36.74|41.45|
> |VCD|13.87|36.92|34.65|41.10|
> |OPERA|16.19|37.58|35.35|41.19|
> |ICD|13.87|32.97|33.49|38.18|
> |MemVR|18.50|36.48|37.67|42.34|
> |MemVR++|18.50|36.48|36.98|42.07|
>
> where the evaluation is conducted on the GPT-4o-mini.
>
> The results demonstrate that MemVR and MemVR++ achieve superior performance on HallusionBench.
>
> > Q8:Minor typos
>
> We have revised our paper accordingly, the typo "preception" is revised to "perception".
>
> We sincerely appreciate your valuable suggestions again.

---

### Official Review · Reviewer_bEEH · 2025-03-13

**Overall Recommendation:** 4

**Summary:**

This paper addresses the hallucination issue in Multimodal Large Language Models (MLLMs) by proposing MemVR, a novel decoding paradigm. MemVR uses visual tokens as supplementary evidence and re-injects them via FFN at the middle trigger layer. Theoretical analysis shows MemVR can mitigate hallucinations by enhancing mutual information, reducing conditional entropy, and optimizing the objective function. Experiments on multiple benchmarks prove its superiority in reducing hallucinations and improving performance.

## update after rebuttal
I have carefully read other reviewers' comments and the rebuttal. Most of the concerns are well addressed. I will keep my original rating as accept.

**Claims And Evidence:**

The claims are well-supported by comprehensive experiments.

**Essential References Not Discussed:**

No.

**Experimental Designs Or Analyses:**

The paper compares MemVR with multiple state-of-the-art methods on various benchmarks. The selection of baselines is comprehensive, covering different types of methods for mitigating hallucinations. The analyses of the experimental results are also valid.

**Methods And Evaluation Criteria:**

By re-injecting visual tokens, MemVR directly addresses the cause of hallucinations in MLLMs, which is the imbalance between visual and textual modalities.

The evaluation criteria, including the use of various benchmark datasets, are appropriate. These datasets cover different aspects of MLLM performance, such as hallucination mitigation and general-purpose capabilities, enabling a comprehensive evaluation of MemVR.

**Other Comments Or Suggestions:**

Typo: Line245 "a input".

Adjust the layout: There are significant distances between the positions of figures and their citations in the main text, which can indeed cause inconvenience to readers. For example, Figure1 is in Page1, and the citation for Figure1 is in Page5 Line232. Similarly, Figure8 is mentioned in Page5 (Line220 right), but Figure8 is in Page8. Moreover, Figure3 (Line198 right) is mentioned after Figure4 (Line199 left) and Figure5 (Line182 right). Please re-plan the page layout of the paper and shorten the distance between the figure and the corresponding citation position in the main text as much as possible.

**Other Strengths And Weaknesses:**

**Strengths:**
1. Originality: MemVR is a highly original approach. It combines the concept of visual retracing inspired by human cognitive behavior with the architecture of MLLMs. This novel combination offers a new perspective on solving the hallucination problem in MLLMs.
2. Experiments: The experimental designs are sound. The paper compares MemVR with multiple state-of-the-art methods on various benchmarks. The selection of baselines is comprehensive, covering different types of methods for mitigating hallucinations. The analyses of the experimental results are also valid.
3. Significance: The research is significant as hallucinations are a major obstacle to the widespread application of MLLMs. MemVR's ability to mitigate hallucinations without sacrificing efficiency can greatly improve the reliability of MLLMs, which is crucial for applications in safety-critical fields such as healthcare and autonomous driving.
4. Clarity: The paper is mostly clear. The concepts, methods, and experimental results are clearly presented. (Minor modifications are needed to further improve its structure, please see below for details).

**Weaknesses:**
1. Missing comparisons with cross-attention based retrieval. In Eq.6 a simple  retrieval process for VR is adopted instead of cross-attention layers as in previous approaches (Li et al., 2022; Alayrac et al., 2022). But no direct experimental comparisons are reported.
2. The organization of the paper, especially the layout can be further improved. For example, in Sec4.3, It would be more clear, if "Static Triggered MemVR" is introduced before "Dynamic Triggered MemVR". Other comments can be found in below Section "Other Comments Or Suggestions".
3. Hyperparameter Tuning: The process of determining the optimal hyperparameters for MemVR, such as the injection ratio of visual information and the strategy for selecting the triggered layers, is complex. This may limit the practical application of MemVR as it requires significant effort to finetune these parameters for different models and tasks.

**Questions For Authors:**

No

**Relation To Broader Scientific Literature:**

The key contribution of MemVR is closely related to the existing literature. Prior works have explored various methods to mitigate hallucinations in MLLMs, such as Retrieval Augmented Generation (RAG), extra fine-tuning, attention intervention, and Contrastive Decoding (CD).

**Theoretical Claims:**

To be honest, I did not carefully check the correctness of the theoretical proofs. The proofs are based on established information theoretic concepts such as mutual information, conditional entropy, and the Data Processing Inequality (DPI).

---

> ### Author Rebuttal · Authors · 2025-03-30
>
> **We sincerely thank Reviewer bEEH for the constructive comments on our work.
> We are very grateful to the reviewer for recognising the novelty of our idea and the richness and rationality of our experiments.**
>
> > Q1: Missing comparisons with cross-attention based retrieval.
>
> Sorry for the confusion, we wish to clarify that the ‘cross-attention’ mentioned in our paper refers specifically to a text–image alignment strategy used during the training, rather than the specific method for mitigating VLM hallucinations. We are simply stating that the VR strategy has less overhead than injecting information through cross-attention. Besides, the cross-attention mechanism needs to be trained.
>
> We make complexity analyses between cross-attention and FFN, and VR operations. Let $d$ be the dimension of the hidden state, $D$ denote the dimension of FFN, and $N_v$ and $N_t$ denote the number of vision/text tokens. We have computational complexity (CO) analysis, CO_cross-attn=$\mathcal{O}((N_v+N_v)N_v d+N_vd^2 )$; CO_FFN=$\mathcal{O}((N_v+N_t) d D)$; CO_VR=$\mathcal{O}((N_v+N_t) d N_v)$. Where CO_VR<  CO_FFN < CO_cross-attn.
>
> > Q2: The layout can be further improved and Typo.
>
> Thanks for your helpful suggestion. We have revised our paper accordingly, including layout and typos, so that it will be easier to read.
>
> > Q3: Hyperparameter Tuning.
>
> We value your concerns. We develop MemVR++, which alters the injection rate of visual information from fixed to self-adaptive, thus discarding the hyperparameter $\alpha$. Specifically, the retracing ratio $\alpha$ is determined by 2*(layer_entropy – entropy_threshold) when MemVR is triggered. This ensures that the higher layer_entropy is, which indicates that the model is more confused, the higher $\alpha$ would be. This is a global strategy for MemVR. We've tested it on several benchmarks, and the evaluation shows that this dynamic retracing strategy is also significantly outperforming the default one. The results are as follows,
>
> LLaVA-Bench:
>
> |Method|Average|all|complex|conv|detail|
> |--------|:--------:|:--------:|:--------:|:--------:|:--------:|
> |LLaVA1.5|64.80|50.80|74.60|52.90|52.10|
> |VCD|63.20|48.50|77.90|52.40|50.80|
> |ICD|56.90|40.20|78.20|35.30|42.20|
> |MemVR|65.17|51.30|77.90|55.90|52.60|
> |MemVR++|65.87|51.70|81.80|51.20|49.60|
>
> POPE (MSCOCO):
>
> |Method|Random|Popular|Adversarial|Average|
> |---|:---:|:---:|:---:|:---:|
> |LLaVA1.5|83.49|79.98|76.03|79.83|
> |VCD|86.84|82.65|77.31|82.27|
> |ICD|84.87|82.93|81.07|82.96|
> |MemVR|88.50|87.10|85.20|86.93|
> |MemVR++|88.40|87.17|85.17|86.92|
>
> POPE (A-OKVQA):
>
> |Method|Random|Popular|Adversarial|Average|
> |---|:---:|:---:|:---:|:---:|
> |LLaVA1.5|83.45|79.90|74.04|79.13|
> |VCD|86.15|81.85|74.97|80.99|
> |ICD|85.57|81.93|77.43|81.64|
> |MemVR|91.10|87.33|80.20|86.21|
> |MemVR++|91.03|87.50|80.23|86.25|
>
> POPE (GQA):
>
> |Method|Random|Popular|Adversarial|Average|
> |---|:---:|:---:|:---:|:---:|
> |LLaVA1.5|83.73|78.17|75.08|78.99|
> |VCD|86.65|80.73|76.09|81.16|
> |ICD|84.90|78.37|75.97|79.75|
> |MemVR|89.60|84.63|81.53|85.25|
> |MemVR++|89.57|84.60|81.57|85.25|
>
> MME Benchmark:
>
> |Method|Overall|Perception|Cognition|
> |--------|:--------:|:--------:|:--------:|
> |LLaVA1.5|1864.68|1508.97|355.71|
> |VCD|1872.87|1515.01|357.86|
> |OPERA|1784.34|1473.62|310.71|
> |ICD|1594.77|1306.91|287.86|
> |MemVR|1896.72|1512.80|383.92|
> |MemVR++|1894.14|1512.00|382.14|
>
> And we supplement the testing of MemVR on HallusionBench.
>
> HallusionBench:
>
> |Method|$fAcc$|$qAcc$|$easy aAcc$|$hard aAcc$|$aAcc$|
> |--------|:--------:|:--------:|:--------:|:--------:|:--------:|
> |LLaVA1.5|17.92|8.13|36.04|36.74|41.45|
> |VCD|13.87|11.43|36.92|34.65|41.10|
> |OPERA|16.19|5.49|37.58|35.35|41.19|
> |ICD|13.87|8.35|32.97|33.49|38.18|
> |MemVR|18.50|9.01|36.48|37.67|42.34|
> |MemVR++|18.50|8.35|36.48|36.98|42.07|
>
> where the evaluation is conducted on the GPT-4o-mini.
>
> The results demonstrate that MemVR and MemVR++ achieve superior performance. We'll include this part in our revised version.
>
> We sincerely thank you for your kind suggestions.

---

### Official Review · Reviewer_2zfj · 2025-03-15

**Overall Recommendation:** 3

**Summary:**

This paper introduces Memory-Space Visual Retracing (MemVR), a novel decoding approach to mitigate hallucinations in Multimodal Large Language Models (MLLMs). The authors posit that hallucinations often occur due to the model's tendency to "forget" visual information during text generation, and they address this by developing a "look-twice" mechanism that reinjects visual tokens into the model's middle layers when uncertainty is detected. Unlike existing contrastive decoding approaches, MemVR modifies intermediate hidden states rather than output logits, avoiding the need for multiple decoding passes. The authors evaluate their method across seven benchmarks and demonstrate superior performance in both hallucination mitigation and general capabilities while maintaining efficiency in inference time.


## update after rebuttal
The authors' rebuttal has addressed my concerns. However, I still doubt the effectiveness of MemVR compared to retrieval-based methods, and I'm concerned that the novelty is somewhat limited, as the reviewer 3FtN mentioned. In my opinion, it's a borderline paper considering the technical contributions.

**Claims And Evidence:**

The paper claims MemVR:
1. mitigates hallucinations more effectively than existing methods；
2. maintains or improves general MLLM capabilities;
3. computationally efficient compared to alternatives;
4. plug-and-play and task-agnostic;

Evidence is provided through extensive evaluations on hallucination benchmarks (POPE, CHAIR) and general benchmarks (MME, MM-Bench, LLaVA-Bench, etc.)

**Essential References Not Discussed:**

The paper covers the most relevant literature.

**Experimental Designs Or Analyses:**

The experimental design is comprehensive, evaluating on:
1. Hallucination-focused benchmarks: POPE (using COCO, A-OKVQA, GQA datasets) and CHAIR
2. General capability benchmarks: MME, MM-Bench, MM-Vet, VizWiz, LLaVA-Bench

Experiments compare MemVR against baseline MLLMs and state-of-the-art hallucination mitigation methods (OPERA, VCD, ICD) using various models (LLaVA-1.5, Qwen-VL).

The authors also conduct ablation studies examining the impact of different injection ratios, uncertainty thresholds, and static vs. dynamic triggering strategies.

**Methods And Evaluation Criteria:**

Yes. This paper adopts several multimodal benchmarks to evaluate the proposed method, such as POPE and CHAIR for evaluating hallucination.

**Other Comments Or Suggestions:**

N/A

**Other Strengths And Weaknesses:**

## Strengths

1. The paper introduces a compelling and intuitive mechanism for hallucination mitigation based on the human cognitive process of "looking twice" when uncertain. The approach of reinjecting visual information at the feature level rather than modifying logits is conceptually different from existing methods.
2. The experiments are extensive, covering both hallucination-specific benchmarks and general capabilities across multiple datasets and model architectures. This demonstrates the method's robustness and general applicability.

## Weaknesses:
1. The paper proposes a "look-twice" mechanism for hallucination mitigation, but the core idea of revisiting visual tokens resembles existing retrieval-based and contrastive decoding approaches. The contribution is incremental rather than fundamentally novel.
2. While the proposed method shows good improvements, a detailed analysis of failure cases could strengthen the paper.
3. The implementation of the "dynamic trigger" mechanism lacks theoretical justification for why specific layers are selected for visual retracing. The selection seems heuristic rather than rigorously derived.
4. While the paper claims that MemVR has minimal computational cost, it still introduces additional FFN operations and dynamic triggering logic. The efficiency advantage over contrastive decoding is clear, but a detailed breakdown of computational overhead is missing.

**Questions For Authors:**

N/A

**Relation To Broader Scientific Literature:**

The authors clearly position MemVR as addressing limitations of contrastive decoding approaches, particularly their inference overhead and potential introduction of noise.

**Theoretical Claims:**

The paper provides a theoretical framework based on information theory:
* MemVR enhances mutual information between hidden states and visual evidence;
* MemVR optimizes the Information Bottleneck objective.

---

> ### Author Rebuttal · Authors · 2025-03-28
>
> Thank you for your insightful comments and kind suggestions.
> ﻿
> > Q1: The contribution of revisiting visual tokens.
>
> MemVR is fundamentally different from the current approaches, you may refer to Table 2 in the paper, where we show comprehensive comparisons with existing methods.
> ﻿
> 1) Retrieval-based approach incorporates knowledge from the external database, which brings about a large memory footprint. *MemVR achieves self-improvement without external knowledge.*
>
> 2) CD-based and attention intervention strategies both bring about high inference latency, due to multiple rounds of inference or the rollback operation. *MemVR mitigates hallucinations and excels beyond SOTA methods without incurring additional time overhead.*
>
> 3) Retrieval-based, CD-based, and attention intervention methods generally act on textual/visual input (i.e., instruction/image), output logits, or attention matrix. *MemVR reinjects visual information at hidden states.*
>
> 4) Compared with other methods, which succeed in hallucination benchmarks but fail on general benchmarks, *MemVR enables constant performance boosting in both hallucination and general benchmarks.*
>
> > Q2: Add analysis of failure cases.
>
> Thank you for your valuable and helpful suggestions. We are keen to explore how reinjecting similar visual features without external data might affect model biases.  To address this, we have collected failure cases from the MME benchmark, in the 'Celebrity,' 'Scene,' and 'Landmark' sub-tasks, where MemVR underperforms compared to the default model, as follows.
> ﻿
> | Right numbers | existence | count | position | color | posters | celebrity | scene | landmark | artwork | OCR | CommR | numerical_cal | translation | code |
> |--- | :----: | :-----: | :----: | :-----: | :-----: | :----: | :----: | :-------: | :----: | :----: | :---: | :---: | :-----: | :-----: |
> |Total| 60| 60 | 60| 60| 294| 340| 400 | 400 | 400| 40  | 140 | 40| 40 | 40  |
> |LLaVA1.5-7B|58| 51|45 | 54| 241| 266| 342 | 352 | 286 | 32 | 97  | 18  | 27  | 21 |
> |MemVR | 58 | 51 | 46| 54| 241| 264| 341| 351 | 288 | 32 | 102 | 18  | 28 | 23|
>
> We categorize MemVR's failures into two types:
>
> 1) Cases where the default model provides the correct answer, but MemVR outputs an incorrect one.
>
> 2) Cases where both the default model and MemVR produce incorrect answers.
>
> For failure type 1, we attribute the failure to the over-disturbance of the default model's reasoning process.  In these instances, the original visual features are sufficient for reasoning, and the reinjected tokens inadvertently disrupt this process, leading to errors.  We are actively investigating methods to mitigate such disturbances.
> For failure type 2, the failures arise from either the excessive complexity of the image or gaps in the LLM's knowledge base, which prevent correct reasoning even after VR.
>
> We will be including the bad cases as well as the analysis in our paper.
>
> > Q3: Why are specific layers selected for VR
>
> The layer selection for the dynamic trigger is theoretically grounded in information entropy analysis. Specifically, we employ layer-wise entropy $H(x)=-\sum p(x) \log p(x)$ as an information bottleneck metric to identify critical transition points where feature uncertainty reaches local maxima, which guides information entropy of the trigger layer from high to low state under the Data Processing Inequality[1]. While our implementation adopts an efficient thresholding mechanism, the core selection criterion stems from established analysis of hierarchical feature stabilization patterns in [2], rather than arbitrary heuristics.
>
> [1] Cover, T. M. et al. Entropy, relative entropy and mutual information. Elements of Information Theory, 2(1):12–13, 1991.
>
> [2] Teerapittayanon et al., BranchyNet: Fast Inference via Early Exiting from Deep Neural Networks, ICPR 2016.
>
> > Q4: detailed breakdown of computational overhead.
>
> LLaVA components include ViT+MLP+LLM (self-attention and FFN). Suppose $L_v$ and $L_l$ are the number of layers of ViT and LLM, $d_v$,$d$, and $D$ are dimensions of ViT, self-attention, and FFN, $N_v$ and $N_t$ denote the number of vision/text tokens.
>
> 1) Computational overhead of VCD: FLOPs$_{\text{VCD}}$=$2*(L_v(N_v^2 d_v+N_v d_v^2)+N_v d_v d+L_l[(N_v+N_t)^2 d+(N_v+N_t) dD])$
>
> 2) Computational overhead of MemVR: FLOPs$_{\text{MemVR}}$=$L_v(N_v^2 d_v+N_v d_v^2)+N_v d_v d+L_l[(N_v+N_t)^2 d+(N_v+N_t) dD] + \underline{(N_v+N_t) d{N_v}+L_o (N_v+N_t) d}$. where $L_o \lt L_l$.
>
> Clearly, MemVR adds $\underline{\text{underline}}$ overhead terms, but it is
> negligible as $(N_v+N_t) d$ is low computation, and  $N_v \ll D$, for instance $D = 11008$ and $N_v = 256$
> for LLaVA, thus $(N_v+N_t) d{N_v}\ll (N_v+N_t) d D$, which makes VR operation efficient in total.
>
> We hope the content above can address your concerns. Please let us know if you have further questions.
>
> We sincerely thank you for your kind suggestions again.

---

### Decision · Program_Chairs · 2025-05-01

**Decision:**

Accept (poster)

**Comment:**

Following the rebuttal, all reviewers have recommended the acceptance of the paper (3 weak accept and 1 accept).

In the initial review, multiple concerns were raised by the reviewers. The AC thoroughly reviewed all of these concerns in conjunction with the rebuttal comments, while also checking the paper itself.
The authors have adequately addressed the majority of the reviewers' concerns.

The primary remaining issue pertains to the significance of the contribution, as highlighted by Reviewers 2zfj and 3FtN. However, the AC finds that the proposed method is sufficiently distinct over the references the reviewers referred to and demonstrates its own clear merits. The AC concludes that the contribution is well-defined, and that the strengths of the paper significantly outweigh the identified weaknesses.

In light of this, the AC concurs with the unanimous recommendation of the reviewers and, therefore, recommends the acceptance of the paper.